# Chebyshev-Cantelli PAC-Bayes-Bennett Inequality for the Weighted Majority Vote

**Yi-Shan Wu**
University of Copenhagen
yswu@di.ku.dk

**Andrés R. Masegosa**
University of Aalborg
arma@cs.aau.dk

**Stephan S. Lorenzen**
University of Copenhagen
lorenzen@di.ku.dk

**Christian Igel**
University of Copenhagen
igel@di.ku.dk

**Yevgeny Seldin**
University of Copenhagen
seldin@di.ku.dk

## Abstract

We present a new second-order oracle bound for the expected risk of a weighted majority vote. The bound is based on a novel parametric form of the Chebyshev-Cantelli inequality (a.k.a. one-sided Chebyshev's), which is amenable to efficient minimization. The new form resolves the optimization challenge faced by prior oracle bounds based on the Chebyshev-Cantelli inequality, the C-bounds [Germain et al., 2015], and, at the same time, it improves on the oracle bound based on second order Markov's inequality introduced by Masegosa et al. [2020]. We also derive a new concentration of measure inequality, which we name PAC-Bayes-Bennett, since it combines PAC-Bayesian bounding with Bennett's inequality. We use it for empirical estimation of the oracle bound. The PAC-Bayes-Bennett inequality improves on the PAC-Bayes-Bernstein inequality of Seldin et al. [2012]. We provide an empirical evaluation demonstrating that the new bounds can improve on the work of Masegosa et al. [2020]. Both the parametric form of the Chebyshev-Cantelli inequality and the PAC-Bayes-Bennett inequality may be of independent interest for the study of concentration of measure in other domains.

## 1 Introduction

Weighted majority vote is a central technique for combining predictions of multiple classifiers. It is an integral part of random forests [Breiman, 1996, 2001], boosting [Freund and Schapire, 1996], gradient boosting [Friedman, 1999, 2001, Mason et al., 1999, Chen and Guestrin, 2016], and it is also used to combine predictions of heterogeneous classifiers. It is part of the winning strategies in many machine learning competitions. The power of the majority vote is in the cancellation of errors effect: when the errors of individual classifiers are independent or anticorrelated and the error probability of individual classifiers is smaller than 0.5, then the errors average out and the majority vote tends to outperform the individual classifiers.

Generalization bounds for weighted majority vote and theoretically-grounded approaches to weight-tuning are decades-old research topics. Berend and Kontorovich [2016] derived an optimal solution under the assumption of known error rates and independence of errors of individual classifiers, but neither of the two assumptions is typically satisfied in practice.

In absence of the independence assumption, the most basic result is the first order oracle bound, which is based on Markov's inequality and bounds the expected loss of $\rho$-weighted majority vote by twice the $\rho$-weighted average of expected losses of the individual classifiers. This finding is so old and basic that Langford and Shawe-Taylor [2002] call it "the folk theorem". The $\rho$-weighted average

35th Conference on Neural Information Processing Systems (NeurIPS 2021).

of the expected losses is then bounded using PAC-Bayesian bounds, turning the oracle bound into an empirical bound [McAllester, 1998, Seeger, 2002, Langford and Shawe-Taylor, 2002]. While the translation from oracle to empirical bound is quite tight [Germain et al., 2009, Thiemann et al., 2017], the first order oracle bound ignores the correlation of errors, which is the main power of the majority vote. As a result, its minimization overconcentrates the weights on the best-performing classifiers, effectively reducing the majority vote to a very few or even just a single best classifier, which leads to a significant deterioration of the test error on held-out data [Lorenzen et al., 2019, Masegosa et al., 2020].

In order to take correlation of errors into account, Lacasse et al. [2007] derived second order oracle bounds, the C-bounds, which are based on the Chebyshev-Cantelli inequality. The ideas were further developed by Laviolette et al. [2011], Germain et al. [2015], and Laviolette et al. [2017]. However, they were only able to optimize the bounds in the highly restrictive setting of binary classification with self-complemented sets of classifiers and aligned priors and posteriors [Germain et al., 2015]. Several follow-up works resorted to minimization of heuristic surrogates rather than the bound itself [Bauvin et al., 2020, Viallard et al., 2021]. Furthermore, second order oracle quantities in the denominator of the oracle bounds lead to looseness in their translation to empirical bounds [Lorenzen et al., 2019].

Masegosa et al. [2020] proposed an alternative second-order oracle bound, the tandem bound, based on second order Markov's inequality. While the second order Markov's inequality is weaker than the Chebyshev-Cantelli inequality, the resulting bound has no oracle quantities in the denominator, which allows tight translation to an empirical bound. Additionally, Masegosa et al. proposed an efficient procedure for minimization of their empirical bound. They have shown that minimization of the empirical bound does not lead to deterioration of the test error.

Our work can be seen as a bridge between the tandem bound and the C-bounds, and as an improvement of both. The key novelty is a new parametric form of Chebyshev-Cantelli inequality, which preserves the tightness of Chebyshev-Cantelli, but avoids oracle quantities in the denominator. This allows both efficient translation to empirical bounds and efficient minimization. We derive two new second order oracle bounds based on the new inequality, one using the tandem loss and the other using the tandem loss with an offset. For empirical estimation of the latter we derive a PAC-Bayes-Bennett inequality. The overall contributions can be summarized as follows:

1. We propose a new parametric form of the Chebyshev-Cantelli inequality, which has no variance in the denominator and preserves tightness of the original bound. The new form allows efficient minimization and empirical estimation.

2. We propose two new second order oracle bounds for the weighted majority vote based on the new form of the Chebyshev-Cantelli inequality. The bounds have two advantages: (1) they are amenable to tight translation to empirical bounds; and (2) the resulting empirical bounds are amenable to efficient minimization.

3. We derive a new concentration of measure inequality, which we name the PAC-Bayes-Bennett inequality. It improves on the PAC-Bayes-Bernstein inequality of Seldin et al. [2012]. We use the inequality for bounding the tandem loss with an offset.

## 2 Problem setup

The problem setup and notations are borrowed from Masegosa et al. [2020].

**Multiclass classification.** We let $S = \{(X_1, Y_1), \ldots, (X_n, Y_n)\}$ be an independent identically distributed sample from $\mathcal{X} \times \mathcal{Y}$, drawn according to an unknown distribution $D$, where $\mathcal{Y}$ is finite and $\mathcal{X}$ is arbitrary. A hypothesis is a function $h : \mathcal{X} \to \mathcal{Y}$, and $\mathcal{H}$ denotes a space of hypotheses. We evaluate the quality of $h$ using the zero-one loss $\ell(h(X), Y) = \mathbb{1}(h(X) \neq Y)$, where $\mathbb{1}(\cdot)$ is the indicator function. The expected loss of $h$ is denoted by $L(h) = \mathbb{E}_{(X,Y) \sim D}[\ell(h(X), Y)]$ and the empirical loss of $h$ on a sample $S$ of size $n$ is denoted by $\hat{L}(h, S) = \frac{1}{n} \sum_{i=1}^{n} \ell(h(X_i), Y_i)$.

**Randomized classifiers.** A *randomized classifier* (a.k.a. Gibbs classifier) associated with a distribution $\rho$ on $\mathcal{H}$, for each input $X$ randomly draws a hypothesis $h \in \mathcal{H}$ according to $\rho$ and predicts $h(X)$. The expected loss of a randomized classifier is given by $\mathbb{E}_{h \sim \rho}[L(h)]$ and the empirical loss by

$\mathbb{E}_{h \sim \rho}[\hat{L}(h, S)]$. To simplify the notation we use $\mathbb{E}_D[\cdot]$ as a shorthand for $\mathbb{E}_{(X,Y) \sim D}[\cdot]$ and $\mathbb{E}_\rho[\cdot]$ as a shorthand for $\mathbb{E}_{h \sim \rho}[\cdot]$.

**Ensemble classifiers and majority vote.** Ensemble classifiers predict by taking a weighted aggregation of predictions by hypotheses from $\mathcal{H}$. The $\rho$-weighted majority vote $\mathrm{MV}_\rho$ predicts $\mathrm{MV}_\rho(X) = \arg\max_{y \in \mathcal{Y}} \mathbb{E}_\rho[\mathbb{1}(h(X) = y)]$, where ties can be resolved arbitrarily.

## 3 A review of prior first and second order oracle bounds

If majority voting makes an error, we know that at least a $\rho$-weighted half of the classifiers have made an error and, therefore, $\ell(\mathrm{MV}_\rho(X), Y) \leq \mathbb{1}(\mathbb{E}_\rho[\mathbb{1}(h(X) \neq Y)] \geq 0.5)$. This observation leads to the well-known first order oracle bound for the loss of weighted majority vote.

**Theorem 1** (First Order Oracle Bound).

$$L(\mathrm{MV}_\rho) \leq 2\mathbb{E}_\rho[L(h)].$$

*Proof.* We have $L(\mathrm{MV}_\rho) = \mathbb{E}_D[\ell(\mathrm{MV}_\rho(X), Y)] \leq \mathbb{P}(\mathbb{E}_\rho[\mathbb{1}(h(X) \neq Y)] \geq 0.5)$. By applying Markov's inequality to random variable $Z = \mathbb{E}_\rho[\mathbb{1}(h(X) \neq Y)]$ we have:

$$L(\mathrm{MV}_\rho) \leq \mathbb{P}(\mathbb{E}_\rho[\mathbb{1}(h(X) \neq Y)] \geq 0.5) \leq 2\mathbb{E}_D[\mathbb{E}_\rho[\mathbb{1}(h(X) \neq Y)]] = 2\mathbb{E}_\rho[L(h)]. \qquad \square$$

PAC-Bayesian analysis can be used to bound $\mathbb{E}_\rho[L(h)]$ in Theorem 1 in terms of $\mathbb{E}_\rho[\hat{L}(h, S)]$, thus turning the oracle bound into an empirical one. The disadvantage of the first order approach is that $\mathbb{E}_\rho[L(h)]$ ignores correlations of predictions, which is the main power of the majority vote.

Masegosa et al. [2020] have used second order Markov's inequality, by which for a non-negative random variable $Z$ and $\varepsilon > 0$

$$\mathbb{P}(Z \geq \varepsilon) = \mathbb{P}(Z^2 \geq \varepsilon^2) \leq \frac{\mathbb{E}[Z^2]}{\varepsilon^2}.$$

For pairs of hypotheses $h$ and $h'$ they have defined the *tandem loss* $\ell(h(X), h'(X), Y) = \mathbb{1}(h(X) \neq Y \wedge h'(X) \neq Y) = \mathbb{1}(h(X) \neq Y)\mathbb{1}(h'(X) \neq Y)$, also termed *joint error* by Lacasse et al. [2007], which counts an error only if both $h$ and $h'$ err on a sample $(X, Y)$. The corresponding *expected tandem loss* is defined by

$$L(h, h') = \mathbb{E}_D[\mathbb{1}(h(X) \neq Y)\mathbb{1}(h'(X) \neq Y)].$$

Lacasse et al. [2007] and Masegosa et al. [2020] have shown that expectation of the second moment of the weighted loss equals expectation of the tandem loss. Using $\rho^2$ as a shorthand for the product distribution $\rho \times \rho$ over $\mathcal{H} \times \mathcal{H}$ and $\mathbb{E}_{\rho^2}[L(h, h')]$ as a shorthand for $\mathbb{E}_{h \sim \rho, h' \sim \rho}[L(h, h')]$, the result is the following.

**Lemma 2** (Masegosa et al., 2020). *In multiclass classification*

$$\mathbb{E}_D[\mathbb{E}_\rho[\mathbb{1}(h(X) \neq Y)]^2] = \mathbb{E}_{\rho^2}[L(h, h')].$$

By combining second order Markov's inequality with Lemma 2, Masegosa et al. have shown the following result.

**Theorem 3** (Masegosa et al., 2020). *In multiclass classification*

$$L(\mathrm{MV}_\rho) \leq 4\mathbb{E}_{\rho^2}[L(h, h')].$$

Lacasse et al. [2007] have used the Chebyshev-Cantelli inequality to derive a different form of a second order oracle bound. We use $\mathbb{V}[Z]$ to denote the variance of a random variable $Z$ in the statement of Chebyshev-Cantelli inequality.

**Theorem 4** (Chebyshev-Cantelli inequality). *For $\varepsilon > 0$*

$$\mathbb{P}(Z - \mathbb{E}[Z] \geq \varepsilon) \leq \frac{\mathbb{V}[Z]}{\varepsilon^2 + \mathbb{V}[Z]}.$$

Theorem 4 together with Lemma 2 leads to the following result, known as the *C-bound*.

**Theorem 5** (Lacasse et al., 2007, Masegosa et al., 2020). *If $\mathbb{E}_\rho[L(h)] \leq \frac{1}{2}$, then*

$$L(\mathrm{MV}_\rho) \leq \frac{\mathbb{E}_{\rho^2}[L(h,h')] - \mathbb{E}_\rho[L(h)]^2}{\frac{1}{4} + \mathbb{E}_{\rho^2}[L(h,h')] - \mathbb{E}_\rho[L(h)]}.$$

Masegosa et al. have shown that the Chebyshev-Cantelli inequality is always at least as tight as second order Markov's inequality (below we provide an alternative and more intuitive proof of this fact) and, therefore, the oracle bound in Theorem 5 is always at least as tight as the oracle bound in Theorem 3. However, the presence of $\mathbb{E}_{\rho^2}[L(h,h')]$ and $\mathbb{E}_\rho[L(h)]$ in the denominator make empirical estimation and optimization of the bound in Theorem 5 impractical, and Theorem 3 was the only practically applicable second order bound so far.

# 4 Main Contributions

We present three main contributions: (1) a new form of the Chebyshev-Cantelli inequality, which is convenient for optimization; (2) an application of the inequality to the analysis of weighted majority vote; and (3) a PAC-Bayes-Bennett inequality, which is used to bound the risk with an offset in the bound for weighted majority vote. We start with the new form of Chebyshev-Cantelli inequality, which can be seen as a refinement of second order Markov's inequality or as an intermediate step in the proof of the Chebyshev-Cantelli inequality.

**Theorem 6.** *For any $\varepsilon > 0$ and all $\mu < \varepsilon$*

$$\mathbb{P}(Z \geq \varepsilon) \leq \frac{\mathbb{E}\left[(Z-\mu)^2\right]}{(\varepsilon-\mu)^2}.$$

*Proof.*

$$\mathbb{P}(Z \geq \varepsilon) = \mathbb{P}(Z - \mu \geq \varepsilon - \mu) \leq \mathbb{P}\big((Z-\mu)^2 \geq (\varepsilon-\mu)^2\big) \leq \frac{\mathbb{E}\left[(Z-\mu)^2\right]}{(\varepsilon-\mu)^2}. \qquad \square$$

The inequality can also be written as

$$\mathbb{P}(Z \geq \varepsilon) \leq \frac{\mathbb{E}\left[(Z-\mu)^2\right]}{(\varepsilon-\mu)^2} = \frac{\mathbb{E}\left[Z^2\right] - 2\mu\mathbb{E}\left[Z\right] + \mu^2}{(\varepsilon-\mu)^2}. \tag{1}$$

The bound is minimized by $\mu^* = \mathbb{E}\left[Z\right] - \frac{\mathbb{V}[Z]}{\varepsilon - \mathbb{E}[Z]}$, which can be verified by taking a derivative of the bound with respect to $\mu$. Note that $\mu^*$ can take negative values. Substitution of $\mu^*$ into the bound and simple algebraic manipulations recover the Chebyshev-Cantelli inequality, whereas $\mu = 0$ recovers second order Markov's inequality. The main advantage of Theorem 6 over the Chebyshev-Cantelli inequality is ease of estimation and optimization due to absence of the variance term in the denominator.

Equation (1) leads to two new second order oracle bounds for the weighted majority vote, given in Theorems 7 and 8.

**Theorem 7.** *In multiclass classification, for all $\rho$ and all $\mu < 0.5$*

$$L(\mathrm{MV}_\rho) \leq \frac{\mathbb{E}_{\rho^2}[L(h,h')] - 2\mu\mathbb{E}_\rho[L(h)] + \mu^2}{(0.5-\mu)^2}.$$

*Proof.* As in the previous section, we take $Z = \mathbb{E}_\rho[\mathbb{1}(h(X) \neq Y)]$, so that $L(\mathrm{MV}_\rho) \leq \mathbb{P}(Z \geq 0.5)$. The result follows by (1) and the calculations of $\mathbb{E}\left[Z^2\right]$ and $\mathbb{E}\left[Z\right]$ from the previous section. Note that the result is a deterministic statement. $\qquad \square$

For $\mu = 0$, Theorem 7 recovers Theorem 3, but if $\mu^* = \mathbb{E}_\rho[L(h)] - \frac{\mathbb{E}_{\rho^2}[L(h,h')] - \mathbb{E}[L(h)]^2}{0.5 - \mathbb{E}_\rho[L(h)]} \neq 0$, then substitution of $\mu^*$ into the theorem yields a tighter oracle bound. At the same time, substitution of $\mu^*$ recovers Theorem 5, but the great advantage of Theorem 7 is that the bound allows easy empirical estimation and optimization with respect to $\rho$, due to absence of $\mathbb{E}_{\rho^2}[L(h,h')]$ and $\mathbb{E}_\rho[L(h)]$ in the

denominator. Thus, Theorem 7 has the oracle tightness of Theorem 5 and the ease of estimation and optimization of Theorem 3. The oracle quantities $\mathbb{E}_{\rho^2}[L(h, h')]$ and $\mathbb{E}_\rho[L(h)]$ can be bounded using PAC-Bayes-kl or PAC-Bayes-$\lambda$ inequalities, as discussed in the next section.

In order to present the second oracle bound we introduce a new quantity. For a pair of hypotheses $h$ and $h'$ and a constant $\mu$, we define *tandem loss with $\mu$-offset*, for brevity $\mu$-*tandem loss*, as

$$\ell_\mu(h(X), h'(X), Y) = (\mathbb{1}(h(X) \neq Y) - \mu)(\mathbb{1}(h'(X) \neq Y) - \mu). \tag{2}$$

Note that it can take negative values. We denote its expectation by

$$L_\mu(h, h') = \mathbb{E}_D[\ell_\mu(h(X), h'(X), Y)] = \mathbb{E}_D[(\mathbb{1}(h(X) \neq Y) - \mu)(\mathbb{1}(h'(X) \neq Y) - \mu)].$$

With $Z = \mathbb{E}_\rho[\mathbb{1}(h(X) \neq Y)]$ as before, we have

$$\mathbb{E}\left[(Z - \mu)^2\right] = \mathbb{E}_D[(\mathbb{E}_\rho[(\mathbb{1}(h(X) \neq Y) - \mu)])^2]$$
$$= \mathbb{E}_{\rho^2}[\mathbb{E}_D[(\mathbb{1}(h(X) \neq Y) - \mu)(\mathbb{1}(h'(X) \neq Y) - \mu)]] = \mathbb{E}_{\rho^2}[L_\mu(h, h')].$$

Now we present our second oracle bound.

**Theorem 8.** *In multiclass classification, for all $\rho$ and all $\mu < 0.5$*

$$L(\mathrm{MV}_\rho) \leq \frac{\mathbb{E}_{\rho^2}[L_\mu(h, h')]}{(0.5 - \mu)^2}.$$

*Proof.* The result follows by Theorem 6 and the calculation above. Note that the inequality is a deterministic statement. □

In order to discuss the advantage of Theorem 8, we define the variance of the $\mu$-tandem loss

$$\mathbb{V}_\mu(h, h') = \mathbb{E}_D[((\mathbb{1}(h(X) \neq Y) - \mu)(\mathbb{1}(h'(X) \neq Y) - \mu) - L_\mu(h, h'))^2].$$

If the variance of the $\mu$-tandem loss is small, we can use Bernstein-type inequalities to obtain tighter estimates compared to kl-type inequalities.

We bound the $\mu$-tandem loss using our next contribution, the PAC-Bayes-Bennett inequality, which improves on the PAC-Bayes-Bernstein inequality derived by Seldin et al. [2012] and may be of independent interest. The inequality holds for any loss function with bounded length of the range, we use $\tilde{\ell}$ and matching tilde-marked quantities to distinguish it from the zero-one loss $\ell$. We let $\tilde{L}(h) = \mathbb{E}_D[\tilde{\ell}(h(X), Y)]$ and $\tilde{\mathbb{V}}(h) = \mathbb{E}_D[(\tilde{\ell}(h(X), Y) - \tilde{L}(h))^2]$ be the expected tilde-loss of $h$ and its variance and let $\hat{\tilde{L}}(h, S) = \frac{1}{n}\sum_{i=1}^n \tilde{\ell}(h(X_i), Y_i)$ be the empirical tilde-loss of $h$ on a sample $S$.

**Theorem 9** (PAC-Bayes-Bennett inequality). *Let $\tilde{\ell}(\cdot, \cdot)$ be an arbitrary loss function taking values in an interval of length $b$, and assume that $\tilde{\mathbb{V}}(h)$ is finite for all $h$. Let $\phi(x) = e^x - x - 1$. Then for any distribution $\pi$ on $\mathcal{H}$ that is independent of $S$ and any $\gamma > 0$ and $\delta \in (0, 1)$, with probability at least $1 - \delta$ over a random draw of $S$, for all distributions $\rho$ on $\mathcal{H}$ simultaneously:*

$$\mathbb{E}_\rho[\tilde{L}(h)] \leq \mathbb{E}_\rho[\hat{\tilde{L}}(h, S)] + \frac{\phi(\gamma b)}{\gamma b^2}\mathbb{E}_\rho[\tilde{\mathbb{V}}(h)] + \frac{\mathrm{KL}(\rho\|\pi) + \ln\frac{1}{\delta}}{\gamma n}.$$

The proof is based on a change of measure argument combined with Bennett's inequality, the details are provided in Appendix A. Note that the result holds for a fixed (but arbitrary) $\gamma > 0$. In case of optimization with respect to $\gamma$ a union bound has to be applied. For a fixed $\rho$ the bound is convex in $\gamma$ and for a fixed $\gamma$ it is convex in $\rho$, although it is not necessarily jointly convex in $\rho$ and $\gamma$. See Appendix D for optimization details. The PAC-Bayes-Bennett inequality is identical to the PAC-Bayes-Bernstein inequality of Seldin et al. [2012, Theorem 7], except that in the latter the coefficient in front of $\mathbb{E}_\rho[\tilde{\mathbb{V}}[h]]$ is $(e - 2)\gamma$ instead of $\frac{\phi(\gamma b)}{\gamma b^2}$. The result improves on the result of Seldin et al. in two ways. First, in the result of Seldin et al. $\gamma$ is restricted to the $(0, 1/b)$ interval, whereas in our result $\gamma$ is unrestricted from above. And second, we can rewrite the coefficient in front of the variance as $\frac{\phi(\gamma b)}{\gamma b^2} = \frac{\phi(\gamma b)}{\gamma^2 b^2}\gamma$, where $\frac{\phi(\gamma b)}{\gamma^2 b^2}$ is a monotonically increasing function of $\gamma$, which in the interval $\gamma \in (0, 1/b)$ satisfies $\lim_{\gamma \to 0}\frac{\phi(\gamma b)}{\gamma^2 b^2} = \frac{1}{2}$ and for $\gamma = 1/b$ it gives $\frac{\phi(\gamma b)}{\gamma^2 b^2} = (e - 2)$. Thus, PAC-Bayes-Bennett is always at least as tight as PAC-Bayes-Bernstein and, at the same time, for $\gamma < 1/b$ it improves the constant coefficient in front of the variance from $(e - 2) \approx 0.72$ down to 0.5 for $\gamma \to 0$. For $\gamma > 1/b$ PAC-Bayes-Bennett also improves on PAC-Bayes-Bernstein, because PAC-Bayes-Bernstein uses the suboptimal value $\gamma = 1/b$ dictated by its restricted range of $\gamma$.

## 5 From oracle to empirical bounds

We obtain empirical bounds on the oracle quantities $\mathbb{E}_{\rho^2}[L(h, h')]$ and $\mathbb{E}_{\rho}[L(h)]$ in Theorem 7 and $\mathbb{E}_{\rho^2}[L_\mu(h, h')]$ in Theorem 8 by using PAC-Bayesian inequalities. The empirical counterpart of the expected tandem loss is the empirical tandem loss

$$\hat{L}(h, h', S) = \frac{1}{n} \sum_{i=1}^{n} \mathbb{1}(h(X_i) \neq Y_i) \mathbb{1}(h'(X_i) \neq Y_i).$$

For bounding $\mathbb{E}_{\rho^2}[L(h, h')]$ and $\mathbb{E}_{\rho}[L(h)]$ we use either PAC-Bayes-kl or PAC-Bayes-$\lambda$ inequalities, both cited below. We use $\mathrm{KL}(\rho \| \pi)$ to denote the Kullback-Leibler divergence between distributions $\rho$ and $\pi$ on $\mathcal{H}$ and $\mathrm{kl}(p \| q)$ to denote the Kullback-Leibler divergence between two Bernoulli distributions with biases $p$ and $q$.

**Theorem 10** (PAC-Bayes-kl Inequality, Seeger, 2002, Maurer, 2004). *For any probability distribution $\pi$ on $\mathcal{H}$ that is independent of $S$ and any $\delta \in (0, 1)$, with probability at least $1 - \delta$ over a random draw of a sample $S$, for all distributions $\rho$ on $\mathcal{H}$ simultaneously:*

$$\mathrm{kl}\left(\mathbb{E}_{\rho}[\hat{L}(h, S)] \middle\| \mathbb{E}_{\rho}[L(h)]\right) \leq \frac{\mathrm{KL}(\rho \| \pi) + \ln(2\sqrt{n}/\delta)}{n}. \tag{3}$$

**Theorem 11** (PAC-Bayes-$\lambda$ Inequality, Thiemann et al., 2017, Masegosa et al., 2020). *For any probability distribution $\pi$ on $\mathcal{H}$ that is independent of $S$ and any $\delta \in (0, 1)$, with probability at least $1 - \delta$ over a random draw of a sample $S$, for all distributions $\rho$ on $\mathcal{H}$ and all $\lambda \in (0, 2)$ and $\gamma > 0$ simultaneously:*

$$\mathbb{E}_{\rho}[L(h)] \leq \frac{\mathbb{E}_{\rho}[\hat{L}(h, S)]}{1 - \frac{\lambda}{2}} + \frac{\mathrm{KL}(\rho \| \pi) + \ln(2\sqrt{n}/\delta)}{\lambda \left(1 - \frac{\lambda}{2}\right) n}, \tag{4}$$

$$\mathbb{E}_{\rho}[L(h)] \geq \left(1 - \frac{\gamma}{2}\right) \mathbb{E}_{\rho}[\hat{L}(h, S)] - \frac{\mathrm{KL}(\rho \| \pi) + \ln(2\sqrt{n}/\delta)}{\gamma n}. \tag{5}$$

(The upper bound (4) is due to Thiemann et al. [2017] and the lower bound (5) is due to Masegosa et al. [2020], and the two bounds hold simultaneously.) The PAC-Bayes-$\lambda$ inequality is an optimization-friendly relaxation of the PAC-Bayes-kl inequality. Therefore, for optimization of $\rho$ we use the PAC-Bayes-$\lambda$ inequality, the upper bound for $\mathbb{E}_{\rho^2}[L(h, h')]$ and the lower or upper bound for $\mathbb{E}_{\rho}[L(h)]$, depending on the positiveness of $\mu$, but once we have converged to a solution we use PAC-Bayes-kl to compute the final bound. The kl form provides both an upper and a lower bound through the upper and lower inverse of the kl.[1] Taking the oracle bound from Theorem 7 and bounding the oracle quantities using Theorem 11 we obtain the following result.

**Theorem 12.** *For any distribution $\pi$ on $\mathcal{H}$ that is independent of $S$, and any $\delta \in (0, 1)$, with probability at least $1 - \delta$ over a random draw of $S$, for all distributions $\rho$ on $\mathcal{H}$, and all $\mu, \lambda,$ and $\gamma$ in the ranges specified below simultaneously, we have:*

- *For $\mu \in [0, 0.5)$, $\lambda \in (0, 2)$, and $\gamma > 0$:*

$$L(\mathrm{MV}_\rho) \leq \frac{1}{(0.5 - \mu)^2} \left[ \frac{\mathbb{E}_{\rho^2}[\hat{L}(h, h', S)]}{1 - \frac{\lambda}{2}} + \frac{2\,\mathrm{KL}(\rho \| \pi) + \ln(4\sqrt{n}/\delta)}{\lambda \left(1 - \frac{\lambda}{2}\right) n} \right.$$
$$\left. - 2\mu \left( \left(1 - \frac{\gamma}{2}\right) \mathbb{E}_{\rho}[\hat{L}(h, S)] - \frac{\mathrm{KL}(\rho \| \pi) + \ln(4\sqrt{n}/\delta)}{\gamma n} \right) + \mu^2 \right].$$

---

[1]Reeb et al. [2018] and Letarte et al. [2019] provide alternative ways of direct minimization of the upper bound on $\mathbb{E}_{\rho}[L(h)]$ given by the upper inverse of kl in the PAC-Bayes-kl bound. We use the PAC-Bayes-$\lambda$ relaxation due to its simplicity, and because it provides an easy way of simultaneous optimization of an upper bound on $\mathbb{E}_{\rho^2}[L(h, h')]$ and a lower or upper bound on $\mathbb{E}_{\rho}[L(h)]$ (depending on $\mu$).

- *For $\mu < 0$, $\lambda \in (0, 2)$, and $\gamma \in (0, 2)$:*

$$L(\mathrm{MV}_\rho) \leq \frac{1}{(0.5 - \mu)^2} \left[ \frac{\mathbb{E}_{\rho^2}[\hat{L}(h, h', S)]}{1 - \frac{\lambda}{2}} + \frac{2\,\mathrm{KL}(\rho\|\pi) + \ln(4\sqrt{n}/\delta)}{\lambda \left(1 - \frac{\lambda}{2}\right) n} \right.$$
$$\left. - 2\mu \left( \frac{\mathbb{E}_\rho[\hat{L}(h, S)]}{1 - \frac{\gamma}{2}} + \frac{\mathrm{KL}(\rho\|\pi) + \ln(4\sqrt{n}/\delta)}{\gamma \left(1 - \frac{\gamma}{2}\right) n} \right) + \mu^2 \right].$$

*Proof.* The result follows by substitution of the upper bound (4) on $\mathbb{E}_{\rho^2}[L(h, h')]$ and the lower bound (5) on $\mathbb{E}_\rho[L(h)]$ in the case of positive $\mu$, or the upper bound (4) on $\mathbb{E}_\rho[L(h)]$ in the case of negative $\mu$, into Theorem 7. We note that $\mathrm{KL}(\rho^2\|\pi^2) = 2\,\mathrm{KL}(\rho\|\pi)$ [Germain et al., 2015, Page 814], which gives the factor 2 in front of the first KL term. The factor 4 in the logarithms comes from a union bound over the bounds on $\mathbb{E}_{\rho^2}[L(h, h')]$ and $\mathbb{E}_\rho[L(h)]$. □

We note that both the loss and the tandem loss are Bernoulli random variables, and for Bernoulli random variables the PAC-Bayes-kl inequality is tighter than the PAC-Bayes-Bennett [Tolstikhin and Seldin, 2013]. However, the empirical counterpart of the expected $\mu$-tandem loss is the empirical $\mu$-tandem loss

$$\hat{L}_\mu(h, h', S) = \frac{1}{n} \sum_{i=1}^{n} (\mathbb{1}(h(X_i) \neq Y_i) - \mu)(\mathbb{1}(h'(X_i) \neq Y_i) - \mu),$$

and the $\mu$-tandem losses are not Bernoulli. Therefore, we use the PAC-Bayes-Bennett inequality, which provides an advantage if the variance of the $\mu$-tandem losses happens to be small. The expected and empirical variance of the $\mu$-tandem losses of a pair of hypotheses $h$ and $h'$ are, respectively, defined by

$$\mathbb{V}_\mu(h, h') = \mathbb{E}_D[((\mathbb{1}(h(X) \neq Y) - \mu)(\mathbb{1}(h'(X) \neq Y) - \mu) - L_\mu(h, h'))^2],$$
$$\hat{\mathbb{V}}_\mu(h, h', S) = \frac{1}{n-1} \sum_{i=1}^{n} \left( (\mathbb{1}(h(X_i) \neq Y_i) - \mu)(\mathbb{1}(h'(X_i) \neq Y_i) - \mu) - \hat{L}_\mu(h, h', S) \right)^2.$$

The empirical variance $\hat{\mathbb{V}}_\mu(h, h', S)$ is an unbiased estimate of $\mathbb{V}_\mu(h, h')$.

Since the PAC-Bayes-Bennett inequality is stated in terms of the oracle variance $\mathbb{E}_\rho[\tilde{\mathbb{V}}(h)]$, we use the result by Tolstikhin and Seldin [2013, Equation (15)] to bound it in terms of the empirical variance. For a general loss function $\tilde{\ell}(\cdot, \cdot)$ (not necessarily within $[0, 1]$), we define the empirical variance of the loss of $h$ by $\hat{\tilde{\mathbb{V}}}(h, S) = \frac{1}{n-1} \sum_{i=1}^{n} (\tilde{\ell}(h(X_i), Y_i) - \hat{\tilde{L}}(h))^2$. We recall that $\tilde{L}$, $\tilde{\mathbb{V}}$, and $\hat{\tilde{L}}$ were defined above Theorem 9. We note that the result of Tolstikhin and Seldin assumes that the losses are bounded in the $[0, 1]$ interval. Rescaling to a general range introduces the squared range factor $c^2$ in front of the last term in the inequality below, since scaling a random variable by $c$ scales the variance by $c^2$.

**Theorem 13** (Tolstikhin and Seldin, 2013). *Let $\tilde{\ell}(\cdot, \cdot)$ be an arbitrary bounded loss function and let $c$ be the length of the loss range. Then for any distribution $\pi$ on $\mathcal{H}$ that is independent of $S$, any $\lambda \in \left(0, \frac{2(n-1)}{n}\right)$, and any $\delta \in (0, 1)$, with probability at least $1 - \delta$ over a random draw of the sample $S$, for all distributions $\rho$ on $\mathcal{H}$ simultaneously:*

$$\mathbb{E}_\rho[\tilde{\mathbb{V}}(h)] \leq \frac{\mathbb{E}_\rho[\hat{\tilde{\mathbb{V}}}(h, S)]}{1 - \frac{\lambda n}{2(n-1)}} + \frac{c^2 \left( \mathrm{KL}(\rho\|\pi) + \ln \frac{1}{\delta} \right)}{n\lambda \left( 1 - \frac{\lambda n}{2(n-1)} \right)}.$$

We note that, similar to the PAC-Bayes-Bennett inequality, but in contrast to the PAC-Bayes-$\lambda$ inequality, the inequality above holds for a fixed value of $\lambda$ and in case of optimization over $\lambda$ a union bound has to be applied.

The last thing that is left is to bound the length of the range of $\mu$-tandem losses defined in equation (2).

**Lemma 14.** *For $\mu < 0.5$ we have that the length of the range of $\ell_\mu(\cdot, \cdot, \cdot)$ is $K_\mu = \max\{1 - \mu, 1 - 2\mu\}$.*

A proof is provided in Appendix B. Taking together the results of Theorems 8, 9, 13, and Lemma 14 we obtain the following result.

**Theorem 15.** *For any parameter grid $\{\gamma_1, \ldots, \gamma_{k_\gamma}\}$ and $\{\lambda_1, \ldots, \lambda_{k_\lambda}\}$, where $\gamma_i > 0$ for all $i$ and $\lambda_i \in \left(0, \frac{2(n-1)}{n}\right)$ for all $i$, any distribution $\pi$ on $\mathcal{H}$ that is independent of $S$, and any $\delta \in (0, 1)$, with probability at least $1 - \delta$ over a random draw of $S$, for all values of $\mu < 0.5$, all distributions $\rho$ on $\mathcal{H}$, and all values of $\gamma$ and $\lambda$ in the parameter grid simultaneously:*

$$L(\mathrm{MV}_\rho) \leq \frac{1}{(0.5 - \mu)^2} \left( \mathbb{E}_{\rho^2}[\hat{L}_\mu(h, h', S)] + \frac{2\,\mathrm{KL}(\rho\|\pi) + \ln\frac{2k_\gamma k_\lambda}{\delta}}{\gamma n} \right.$$

$$\left. + \frac{\phi(\gamma K_\mu)}{\gamma K_\mu^2} \left( \frac{\mathbb{E}_{\rho^2}[\hat{\mathbb{V}}_\mu(h, h', S)]}{1 - \frac{\lambda n}{2(n-1)}} + \frac{K_\mu^2 \left(2\,\mathrm{KL}(\rho\|\pi) + \ln\frac{2k_\gamma k_\lambda}{\delta}\right)}{n\lambda\left(1 - \frac{\lambda n}{2(n-1)}\right)} \right) \right).$$

*Proof.* The result follows by reverse substitution of the result of Lemma 14 into Theorem 13, then into Theorem 9, and finally into Theorem 8. Since $\mathrm{KL}(\rho^2\|\pi^2) = 2\,\mathrm{KL}(\rho\|\pi)$, we have factor 2 in front of the KL terms. The factor $2k_\gamma k_\lambda$ comes from a union bound over the parameter grid and the bounds in Theorems 9 and 13. □

## 6 Experiments

We start with a simulated comparison of the oracle bounds and then present an empirical evaluation on real data. The python source code for replicating the experiments is available at Github[2].

**Comparison of the oracle bounds**

Figure 1 depicts a comparison of the second order oracle bound based on the Chebyshev-Cantelli inequality (Theorems 5, 7 and 8, which, as oracle bounds, are equivalent) and the second order oracle bound based on the second order Markov's inequality (Theorem 3). We plot the ratio of the right hand side of the bound in Theorem 7 for the optimal value $\mu^* = \mathbb{E}_\rho[L(h)] - \frac{\mathbb{E}_{\rho^2}[L(h,h')] - \mathbb{E}_\rho[L(h)]^2}{0.5 - \mathbb{E}_\rho[L(h)]}$ to the value of the right hand side of the bound in Theorem 3. A simple calculation shows that if $\mathbb{E}_{\rho^2}[L(h, h')] = 0.5\mathbb{E}_\rho[L(h)]$, then $\mu^* = 0$, which recovers the bound in Theorem 3. The line $\mathbb{E}_{\rho^2}[L(h, h')] = 0.5\mathbb{E}_\rho[L(h)]$ is shown in black in Figure 1. We also note that $\mathbb{E}_\rho[L(h)]^2 \leq \mathbb{E}_{\rho^2}[L(h, h')] \leq \mathbb{E}_\rho[L(h)]$, which defines the feasible region in Figure 1. Whenever $\mathbb{E}_{\rho^2}[L(h, h')] \neq 0.5\mathbb{E}_\rho[L(h)]$ the Chebyshev-Cantelli inequality provides an improvement over second order Markov's inequality. The region above the black line, where $\mathbb{E}_{\rho^2}[L(h, h')] > 0.5\mathbb{E}_\rho[L(h)]$, is the region of high correlation of errors and in this case majority vote

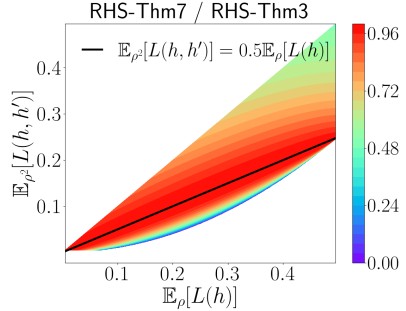

Figure 1: Theorem 7 vs. Theorem 3

yields little improvement over individual classifiers. In this region the first order oracle bound is tighter than the second order oracle bounds (see Appendix C). The region below the black line, where $\mathbb{E}_{\rho^2}[L(h, h')] < 0.5\mathbb{E}_\rho[L(h)]$, is the region of low correlation of errors. In this region the second order oracle bounds are tighter than the first order oracle bound. Note that the potential for improvement below the black line is much higher than above it.

**Empirical evaluation on real datasets**

We studied the empirical performance of the bounds using standard random forest [Breiman, 2001] and a combination of heterogeneous classifiers on a subset of data sets from UCI and LibSVM

---

[2]`https://github.com/StephanLorenzen/MajorityVoteBounds`

repositories [Dua and Graff, 2019, Chang and Lin, 2011]. An overview of the data sets is given in Appendix E.1. The number of points varied from 3000 to 70000 with dimensions $d < 1000$. For each data set, we set aside 20% of the data for the test set $S_{\text{test}}$ and used the remaining data $S$ for ensemble construction, weight optimization and bound evaluation. We evaluate the classifiers and bounds obtained by minimizing the tandem bound TND [Masegosa et al., 2020, Theorem 9], which is the empirical bound on the oracle tandem bound in Theorem 3, the Chebyshev-Cantelli bound with TND empirical loss estimate bound CCTND (Theorem 12), and the Chebyshev-Cantelli bound with PAC-Bayes-Bennett loss estimate bound CCPBB (Theorem 15). We made 10 repetitions of each experiment.

**Ensemble construction and minimization of the bounds.** We follow the construction used by Masegosa et al. [2020]. The idea is to generate multiple random splits of the data set $S$ into pairs of subsets $S = T_h \cup S_h$, such that $T_h \cap S_h = \emptyset$. Each hypothesis is trained on $T_h$ and the empirical loss on $S_h$ provides an unbiased estimate of its expected loss. Note that the splits cannot depend on the data. For our experiments, we generate these splits by bagging, where out-of-bag (OOB) samples $S_h$ provide unbiased estimates of expected losses of individual hypotheses $h$. The resulting set of hypotheses produces an ensemble. As in the work of Masegosa et al., two modifications are required to apply the bounds: the empirical losses $\hat{L}(h, S)$ in the bounds are replaced by the validation losses $\hat{L}(h, S_h)$, and the sample size $n$ is replaced by the minimal validation size $\min_h |S_h|$. For pairs of hypotheses $(h, h')$, we take the overlaps of their validation sets $S_h \cap S_{h'}$ to calculate an unbiased estimate of their tandem loss $\hat{L}(h, h', S_h \cap S_{h'})$, $\mu$-tandem loss $\hat{L}_\mu(h, h', S_h \cap S_{h'})$, and the variance of the $\mu$-tandem loss $\hat{\mathbb{V}}_\mu(h, h', S_h \cap S_{h'})$, which replaces the corresponding empirical losses in the bounds. The sample size is then replaced by $\min_{h,h'} |S_h \cap S_{h'}|$. The details on bound minimization are provided in Appendix D.

**Optimizing weighted random forest.** In the first experiment we compare TND, CCTND, and CCPBB bounds in the setting studied by Masegosa et al. [2020]. We take 100 fully grown trees, use the Gini criterion for splitting, and consider $\sqrt{d}$ features in each split. Figure 2a compares the loss of the random forest on $S_{\text{test}}$ using either uniform weighting $\rho_u$ or optimized weighting $\rho^*$ found by minimization of the three bounds (we exclude the first order bound from the comparison, since it was shown by Masegosa et al. that it significantly deteriorates the test error of the ensemble). While CCTND often performs similar to TND, we find that optimizing using CCPBB often improves accuracy. Figure 2b compares the tightness of the optimized CCTND and CCPBB bounds to the optimized TND bound. The CCTND is generally comparable to TND, while CCPBB is consistently looser than TND, mainly due to the union bounds. The numerical values for the losses and the bounds can be found in Tables 2 and 3 in Appendix E.2.

**Ensembles with heterogeneous classifiers.** In the second experiment, we consider ensembles of heterogeneous classifiers (Linear Discriminant Analysis, $k$-Nearest Neighbors, Decision Tree, Logistic Regression, and Gaussian Naive Bayes). A detailed description is provided in Appendix E.3. Compared to random forests, the variation in performance of ensemble members is larger here. Figure 2c compares the ratio of the loss of the majority vote with optimized weighting to the loss of majority vote with uniform weighting on $S_{\text{test}}$ for $\rho^*$ found by minimization of the first order bound (FO), TND, CCTND, and CCPBB. The numerical values are given in Table 5 in Appendix E.3. We observed that optimizing the FO tends to improve the ensemble accuracy in some cases but degrade in others. However, TND, CCTND, and CCPBB almost always improve the performance w.r.t. the uniform weighting. Table 5 also shows that choosing the best single hypothesis gives almost identical results as optimizing FO. Figure 2d compares the tightness of the CCTND and CCPBB bounds relative to the TND bound. The numerical values are given in Table 6 in Appendix E.3. In this case, we have that CCTND is usually tighter than TND, while CCPBB is usually looser than TND due to the union bounds.

## 7 Discussion

We derived an optimization-friendly form of the Chebyshev-Cantelli inequality and applied it to derive two new forms of second order oracle bounds for the weighted majority vote. The new oracle bounds bridge between the C-bounds [Germain et al., 2015] and the tandem bound [Masegosa et al., 2020]

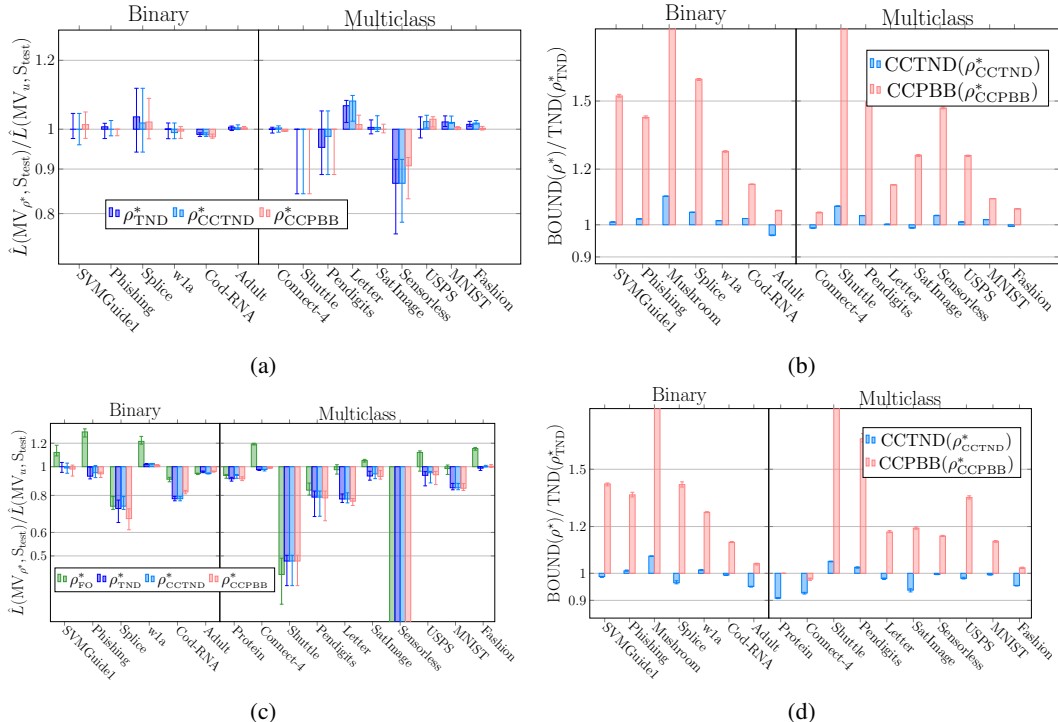

Figure 2: **(a,b) Optimized weighted random forest. (c,d) Ensembles with heterogeneous classifiers.** The median, 25%, and 75% quantiles of: (a,c) the ratio $\hat{L}(\mathrm{MV}_{\rho^*}, \mathrm{S}_{\text{test}})/\hat{L}(\mathrm{MV}_u, \mathrm{S}_{\text{test}})$ of the test loss of the majority vote with optimized weighting $\rho^*$ generated by TND, CCTND and CCPBB to the test loss of majority vote with uniform weighting, and (b,d) the ratio $\mathrm{BOUND}(\rho^\star)/\mathrm{TND}(\rho^\star_{\text{TND}})$ of the CCTND and CCPBB bounds to the TND bound with the corresponding optimized weighting . The plots are on a logarithmic scale. Values above 1 represent degradation and values below 1 represent improvement. Data sets with $L(\mathrm{MV}_u, \mathrm{S}_{\text{test}}) = 0$ are left out in (a,c).

and take the best of both: the tightness of the Chebyshev-Cantelli inequality and the optimization and estimation convenience of the tandem bound. We also derived the PAC-Bayes-Bennett inequality, improving on the PAC-Bayes-Bernstein inequality of Seldin et al. [2012].

Our paper opens several directions for future research. One of them is a better treatment of parameter search in parametric bounds that would give tighter bounds than a union bound over a grid. It would also be interesting to find other applications for the new form of Chebyshev-Cantelli inequality and the PAC-Bayes-Bennett inequality.

## Acknowledgments and Disclosure of Funding

We thank the anonymous reviewers, as well as Tim van Erven, Wouter Koolen, and Peter Grünwald for their constructive feedback, references, and for pointing out that there is no need in a union bound over a grid of $\mu$ in Theorems 12 and 15, and that negative $\mu$ in Theorem 12 requires a separate treatment.

This project has received funding from European Union's Horizon 2020 research and innovation programme under the Marie Skłodowska-Curie grant agreement No 801199. YW and YS acknowledge support by the Independent Research Fund Denmark, grant number 0135-00259B. SSL acknowledges funding by the Danish Ministry of Education and Science, Digital Pilot Hub and Skylab Digital. CI acknowledges support by the Villum Foundation through the project Deep Learning and Remote Sensing for Unlocking Global Ecosystem Resource Dynamics (DeReEco). AM is funded by the Spanish Ministry of Science, Innovation and Universities and by the regional government of Andalucía, grant numbers PID2019-106758GB-C32 and P20-00091, respectively, and by FEDER funds.

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
