## A  A proof of the PAC-Bayes-Bennett inequality (Theorem 9) and a comparison with the PAC-Bayes-Bernstein inequality

In this section we provide a proof of Theorem 9 and a numerical comparison with the PAC-Bayes-Bernstein inequality. The proof is based on the standard change of measure argument. We use the following version by Tolstikhin and Seldin [2013].

**Lemma 16** (PAC-Bayes Lemma). *For any function $f_n : \mathcal{H} \times (\mathcal{X} \times \mathcal{Y})^n \to \mathbb{R}$ and for any distribution $\pi$ on $\mathcal{H}$, such that $\pi$ is independent of $S$, with probability at least $1 - \delta$ over a random draw of $S$, for all distributions $\rho$ on $\mathcal{H}$ simultaneously:*

$$\mathbb{E}_\rho[f_n(h, S)] \leq \mathrm{KL}(\rho\|\pi) + \ln \frac{1}{\delta} + \ln \mathbb{E}_\pi[\mathbb{E}_{S'}[e^{f_n(h, S')}]].$$

The second ingredient is Bennett's lemma, which is a bound on the moment generating function used in the proof of Bennett's inequality. Since we are unaware of a reference, we provide a proof below, which is essentially an intermediate step in the proof of Bennett's inequality [Boucheron et al., 2013, Theorem 2.9].

**Lemma 17** (Bennett's Lemma). *Let $b > 0$ and let $Z_1, \ldots, Z_n$ be i.i.d. zero-mean random variables with finite variance, such that $Z_i \leq b$ for all $i$. Let $M_n = \sum_{i=1}^n Z_i$ and $V_n = \sum_{i=1}^n \mathbb{E}\left[Z_i^2\right]$. Let $\phi(u) = e^u - u - 1$. Then for any $\lambda > 0$:*

$$\mathbb{E}\left[e^{\lambda M_n - \frac{\phi(b\lambda)}{b^2} V_n}\right] \leq 1.$$

*Proof.* Since $u^{-2}\phi(u)$ is a non-decreasing function of $u \in \mathbb{R}$ (where at zero we continuously extend the function), for all $i \in [n]$ and $\lambda > 0$ we have

$$e^{\lambda Z_i} - \lambda Z_i - 1 \leq Z_i^2 \frac{\phi(b\lambda)}{b^2},$$

which implies

$$\mathbb{E}\left[e^{\lambda Z_i}\right] \leq 1 + \lambda \mathbb{E}\left[Z_i\right] + \frac{\phi(b\lambda)}{b^2}\mathbb{E}\left[Z_i^2\right] \leq e^{\frac{\phi(b\lambda)}{b^2}\mathbb{E}[Z_i^2]},$$

where the second inequality uses the assumption that $\mathbb{E}\left[Z_i\right] = 0$ and the fact that $1 + x \leq e^x$ for all $x \in \mathbb{R}$. By the above inequality and independence of the random variables,

$$\mathbb{E}\left[e^{\lambda M_n - \frac{\phi(b\lambda)}{b^2} V_n}\right] = \mathbb{E}\left[\prod_{i=1}^n e^{\lambda Z_i - \frac{\phi(b\lambda)}{b^2}\mathbb{E}[Z_i^2]}\right] = \prod_{i=1}^n \mathbb{E}\left[e^{\lambda Z_i - \frac{\phi(b\lambda)}{b^2}\mathbb{E}[Z_i^2]}\right] \leq 1.$$

$\square$

Now we are ready to prove the theorem.

*Proof of Theorem 9.* We take $f_n(h, S) = \gamma n \left(\tilde{L}(h) - \hat{\tilde{L}}(h, S)\right) - \frac{\phi(\gamma b)}{b^2} n \tilde{\mathbb{V}}(h)$. Then by Lemma 17 we have $\mathbb{E}_S[e^{f_n(h, S)}] \leq 1$. By plugging this into Lemma 16, normalizing by $\gamma n$, and changing sides, we obtain the result. $\square$

### Numerical comparison of the PAC-Bayes-Bennett and PAC-Bayes-Bernstein bound

Figure 3 provides a numerical comparison of the PAC-Bayes-Bennett and PAC-Bayes-Bernstein inequalities (Theorem 9 and Theorem 7 by Tolstikhin and Seldin [2013]).

## B  Proof of Lemma 14

*Proof.* Recall that

$$\ell_\mu(h(X), h'(X), Y) = (\mathbb{1}(h(X) \neq Y) - \mu)(\mathbb{1}(h'(X) \neq Y) - \mu) \in \left\{(1 - \mu)^2, -\mu(1 - \mu), \mu^2\right\}.$$

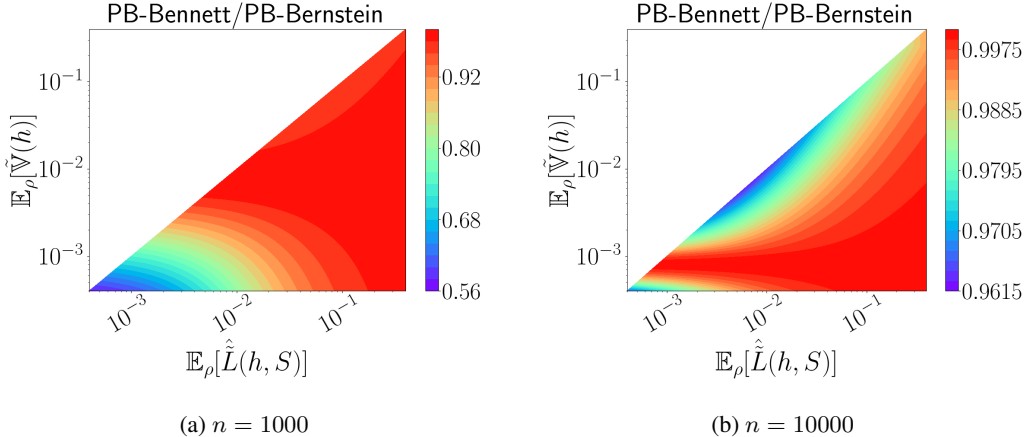

(a) $n = 1000$          (b) $n = 10000$

Figure 3: The ratio of PAC-Bayes Bennett to PAC-Bayes Bernstein bound as a function of $\mathbb{E}_\rho[\hat{\tilde{L}}(h,S)]$ and $\mathbb{E}_\rho[\tilde{\mathbb{V}}(h)]$. We set $\mathrm{KL}(\rho\|\pi) = 5$ and $\delta = 0.05$. The value of $n$ is provided in the captions of the subfigures.

For $\mu < 0.5$, we have $-\mu(1-\mu) < (1-\mu)^2$ and $\mu^2 < (1-\mu)^2$. Therefore, $\ell_\mu(h(X), h'(X), Y) \leq (1-\mu)^2$.

Furthermore, for $\mu < 0$ we have $\mu^2 < -\mu(1-\mu)$, and for $\mu > 0$ we have $-\mu(1-\mu) \leq \mu^2$. Therefore, for $\mu < 0.5$ we have $\ell_\mu(h(X), h'(X), Y) \geq \min\{-\mu(1-\mu), \mu^2\}$.

By combining the upper and the lower bound, we obtain

$$\begin{aligned} K_\mu &= (1-\mu)^2 - \min\{-\mu(1-\mu), \mu^2\} \\ &= \max\{(1-\mu)^2 - (-\mu(1-\mu)), (1-\mu)^2 - \mu^2\} \\ &= \max\{1-\mu, 1-2\mu\}. \end{aligned}$$

□

## C    Comparison of the first and second order oracle bounds

In this section we show that if $\mathbb{E}_\rho[L(h)] < 0.5$ and $\mathbb{E}_{\rho^2}[L(h,h')] > 0.5\mathbb{E}_\rho[L(h)]$, then the first order oracle bound is tighter than the second order oracle bounds, and if $\mathbb{E}_\rho[L(h)] < 0.5$ and $\mathbb{E}_{\rho^2}[L(h,h')] < 0.5\mathbb{E}_\rho[L(h)]$, then it is the other way around.

For comparison of the first order oracle bound $L(\mathrm{MV}_\rho) \leq 2\mathbb{E}_\rho[L(h)]$ vs. the second order oracle tandem bound $L(\mathrm{MV}_\rho) \leq 4\mathbb{E}_{\rho^2}[L(h,h')]$ the statement above is evident.

For the second order oracle bounds based on the Chebyshev-Cantelli inequality we have

$$\frac{\mathbb{E}_{\rho^2}[L(h,h')] - \mathbb{E}_\rho[L(h)]^2}{0.25 + \mathbb{E}_{\rho^2}[L(h,h')] - \mathbb{E}_\rho[L(h)]} \quad \text{vs.} \quad 2\mathbb{E}_\rho[L(h)],$$

$$\mathbb{E}_{\rho^2}[L(h,h')] - \mathbb{E}_\rho[L(h)]^2 \quad \text{vs.} \quad 0.5\mathbb{E}_\rho[L(h)] + 2\mathbb{E}_\rho[L(h)]\mathbb{E}_{\rho^2}[L(h,h')] - 2\mathbb{E}_\rho[L(h)]^2,$$

$$\mathbb{E}_{\rho^2}[L(h,h')](1 - 2\mathbb{E}_\rho[L(h)]) \quad \text{vs.} \quad 0.5\mathbb{E}_\rho[L(h)](1 - 2\mathbb{E}_\rho[L(h)]),$$

$$\mathbb{E}_{\rho^2}[L(h,h')] \quad \text{vs.} \quad 0.5\mathbb{E}_\rho[L(h)],$$

where under the assumption that $\mathbb{E}_\rho[L(h)] < 0.5$ we can cancel $(1 - 2\mathbb{E}_\rho[L(h)])$, since it is positive, and the result is again evident.

## D    Minimization of the bounds

In this section we provide technical details on minimization of the bounds in Theorems 12 and 15. As most of the other PAC-Bayesian works, we take $\pi$ to be a union distribution over the hypotheses

in both cases. As discussed in Section 6, we build a set of data-dependent hypotheses by splitting the data set $S$ into pairs of subsets $S = T_h \cup S_h$, such that $T_h \cap S_h = \emptyset$, training $h$ on $T_h$ and calculating an unbiased loss estimate $\hat{L}(h, S_h)$ on $S_h$. For tandem losses we compute the unbiased estimates $\hat{L}(h, h', S_h \cap S_{h'})$ on the intersections of the corresponding sets $S_h$ and $S_{h'}$.

## D.1 Minimization of the bound in Theorem 12

The adjustment of the bound from Theorem 12 to this construction is for $\mu \geq 0$:

$$
L(\mathrm{MV}_\rho) \leq \frac{1}{(0.5 - \mu)^2} \left[ \frac{\mathbb{E}_{\rho^2}[\hat{L}(h, h', S_h \cap S_{h'})]}{1 - \frac{\lambda}{2}} + \frac{2 \, \mathrm{KL}(\rho \| \pi) + \ln(4\sqrt{m}/\delta)}{\lambda \left(1 - \frac{\lambda}{2}\right) m} \right.
$$
$$
\left. - 2\mu \left( \left(1 - \frac{\gamma}{2}\right) \mathbb{E}_\rho[\hat{L}(h, S_h)] - \frac{\mathrm{KL}(\rho \| \pi) + \ln(4\sqrt{n}/\delta)}{\gamma n} \right) + \mu^2 \right],
$$

and for $\mu < 0$:

$$
L(\mathrm{MV}_\rho) \leq \frac{1}{(0.5 - \mu)^2} \left[ \frac{\mathbb{E}_{\rho^2}[\hat{L}(h, h', S_h \cap S_{h'})]}{1 - \frac{\lambda}{2}} + \frac{2 \, \mathrm{KL}(\rho \| \pi) + \ln(4\sqrt{m}/\delta)}{\lambda \left(1 - \frac{\lambda}{2}\right) m} \right.
$$
$$
\left. - 2\mu \left( \frac{\mathbb{E}_\rho[\hat{L}(h, S_h)]}{1 - \frac{\gamma}{2}} + \frac{\mathrm{KL}(\rho \| \pi) + \ln(4\sqrt{n}/\delta)}{\gamma \left(1 - \frac{\gamma}{2}\right) n} \right) + \mu^2 \right],
$$

where $m = \min_{h,h'} |S_h \cap S_{h'}|$ and $n = \min_h |S_h|$. Below we provide the pseudocode and derive update rules for $\mu$, $\lambda$, $\gamma$, and $\rho$ for alternating minimization of this bound.

---

**Algorithm 1:** Minimization of the bound in Theorem 12

---

**Input:** $m, n$, tandem losses $\hat{L}(h, h', S_h \cap S_{h'})$ for all $h, h'$, and Gibbs losses $\hat{L}(h, S_h)$ for all $h$

**Initialize:** $\rho = \pi$ and $\mu = 0$

**while** The improvement of the bound is larger than $10^{-9}$ **do**

    Compute $\lambda_\rho^*$, the optimal $\lambda$ given $\rho$

    Compute $\gamma_\rho^*$, the optimal $\gamma$ given $\rho$ and $\mu$

    Compute the bound using $\rho$, $\mu$, $\lambda_\rho^*$ and $\gamma_\rho^*$

    Compute new $\mu_\rho^*$, the optimal $\mu$ given $\rho$, $\lambda_\rho^*$ and $\gamma_\rho^*$

    Update the new distribution $\rho'$ with gradient descent given $\mu$, $\lambda_\rho^*$ and $\gamma_\rho^*$

    Let $\rho = \rho'$ and $\mu = \mu_\rho^*$

**end while**

---

**Optimal $\lambda$ given $\rho$**    Minimization of the bound with respect to $\lambda$ is identical to minimization of the tandem bound by Masegosa et al. [2020, Theorem 9]. Masegosa et al. derive the optimal value of $\lambda$:

$$
\lambda_\rho^* = \frac{2}{\sqrt{\frac{2m\mathbb{E}_{\rho^2}[\hat{L}(h, h', S_h \cap S_{h'})]}{2 \, \mathrm{KL}(\rho || \pi) + \ln \frac{4\sqrt{m}}{\delta}} + 1} + 1}.
$$

**Optimal $\gamma$ given $\rho$ and $\mu$**    Minimization of the bound with respect to $\gamma$ in the case of $\mu \geq 0$ is analogous to minimization of the bound by Masegosa et al. [2020, Theorem 10] with respect to $\gamma$. Masegosa et al. derive the optimal value of $\gamma$:

$$
\gamma_\rho^* = \sqrt{\frac{2 \, \mathrm{KL}(\rho || \pi) + \ln(16n/\delta^2)}{n\mathbb{E}_\rho[\hat{L}(h, S_h)]}}.
$$

On the other hand, the optimal $\gamma$ in the case of $\mu < 0$ is analogous to the optimal $\lambda$ above:

$$
\gamma_\rho^* = \frac{2}{\sqrt{\frac{2n\mathbb{E}_\rho[\hat{L}(h, S_h)]}{\mathrm{KL}(\rho || \pi) + \ln \frac{4\sqrt{n}}{\delta}} + 1} + 1}.
$$

**Optimal $\mu$ given $\rho$**   Given $\rho$, we can compute the optimal $\lambda_\rho^*$ and $\gamma_\rho^*$ by the above formulas. Let

$$U_T(\rho) := \frac{\mathbb{E}_{\rho^2}[\hat{L}(h, h', S_h \cap S_{h'})]}{1 - \frac{\lambda_\rho^*}{2}} + \frac{2\,\mathrm{KL}(\rho\|\pi) + \ln(4\sqrt{m}/\delta)}{\lambda_\rho^* \left(1 - \frac{\lambda_\rho^*}{2}\right) m},$$

$$L_G(\rho) := \begin{cases} \left(1 - \frac{\gamma_\rho^*}{2}\right) \mathbb{E}_\rho[\hat{L}(h, S_h)] - \frac{\mathrm{KL}(\rho\|\pi) + \ln(4\sqrt{n}/\delta)}{\gamma_\rho^* n}, & \mu \geq 0 \\ \frac{\mathbb{E}_\rho[\hat{L}(h, S_h)]}{1 - \frac{\gamma_\rho^*}{2}} + \frac{\mathrm{KL}(\rho\|\pi) + \ln(4\sqrt{n}/\delta)}{\gamma_\rho^* \left(1 - \frac{\gamma_\rho^*}{2}\right) n}, & \mu < 0 \end{cases}$$

Then the optimal $\mu$ is

$$\mu_\rho^* = \frac{\frac{1}{2} L_G(\rho) - U_T(\rho)}{\frac{1}{2} - L_G(\rho)}.$$

**Gradient w.r.t. $\rho$ given $\lambda$, $\gamma$ and $\mu$**   Minimization of the bound w.r.t. $\rho$ is equivalent to constrained optimization of $f(\rho) = a\mathbb{E}_{\rho^2}[\hat{L}(h, h', S_h \cap S_{h'})] - 2b\mathbb{E}_\rho[\hat{L}(h, S_h)] + 2c\,\mathrm{KL}(\rho\|\pi)$, where for $\mu \geq 0$, $a = 1/(1 - \lambda/2)$, $b = \mu(1 - \gamma/2)$ and $c = 1/(\lambda(1 - \lambda/2)m) + \mu/(\gamma n)$, and for $\mu < 0$, $a = 1/(1 - \lambda/2)$, $b = \mu/(1 - \gamma/2)$, and $c = 1/(\lambda(1 - \lambda/2)m) - \mu/(\gamma(1 - \gamma/2)n)$. The constraint is that $\rho$ is a probability distribution. We optimize $\rho$ by projected gradient descent, where we iteratively take steps in the direction of the negative gradient of $f$ and project the result onto the probability simplex.

We use $\hat{L}$ to denote the vector of empirical losses and $\hat{L}_{\mathrm{tnd}}$ to denote the matrix of tandem losses. Let $\nabla f$ denote the gradient of $f$ w.r.t. $\rho$ and $(\nabla f)_h$ the $h$-th coordinate of the gradient. We have:

$$(\nabla f)_h = 2\left(a \sum_{h'} \rho(h')\hat{L}(h, h', S_h \cap S_{h'}) - b\hat{L}(h, S_h) + c\left(1 + \ln\frac{\rho(h)}{\pi(h)}\right)\right),$$

$$\nabla f = 2\left(a\hat{L}_{\mathrm{tnd}}\rho - b\hat{L} + c\left(1 + \ln\frac{\rho}{\pi}\right)\right).$$

**Gradient descent optimization w.r.t. $\rho$**   To optimize the weighting $\rho$, we applied iRProp+ for the gradient based optimization, a first order method with adaptive individual step sizes [Igel and Hüsken, 2003, Florescu and Igel, 2018], until the bound did not improve for 10 iterations.

## D.2   Minimization of the bound in Theorem 15

We start with the details on construction of the grid of $\mu$, $\lambda$ and $\gamma$.

### D.2.1   The $\mu$ grid for Theorem 15

We were unable to find a closed-form solution for minimization of the bound w.r.t. $\mu$ and applied a heuristic. Empirically we observed that the bound was quasiconvex in $\mu$ (we were unable to prove that it is always the case) and applied binary search for $\mu$ in the grid. Note that even if we take a grid of $\mu$, we don't need a union bound since the bound holds with high probability for all $\mu$ simultaneously.

We then consider the relevant range of $\mu$. By Theorem 6, we have $\mu < 0.5$. At the same time, $\mu^* = \frac{0.5\mathbb{E}_\rho[L(h)] - \mathbb{E}_{\rho^2}[L(h,h)]}{0.5 - \mathbb{E}_\rho[L(h)]}$, and in Section 6 we have shown that the primary region of interest is where $\mathbb{E}_{\rho^2}[L(h, h')] < 0.5\mathbb{E}_\rho[L(h)]$, which corresponds to $\mu^* > 0$. However, since $\mathbb{E}_{\rho^2}[L(h, h)]$ and $\mathbb{E}_\rho[L(h)]$ are unobserved and we use an upper bound for the first and a lower bound for the second instead, we take a broader range of $\mu$. By making a mild assumption that the upper bound for the tandem loss $\mathbb{E}_{\rho^2}[L(h, h')]$ is at most 0.25 and the lower bound for the Gibbs loss $\mathbb{E}_\rho[L(h)]$ is at most 0.5, we have $\mu \in [-0.5, 0.5)$. We take 400 uniformly spaced points in the selected range for the CCPBB bound.

### D.2.2   The $\lambda$ grid for Theorem 15

The parameter $\lambda$ comes from Theorem 13. The theorem is identical to the result by Tolstikhin and Seldin [2013, Equation (15)], except rescaling, but rescaling happens on top of the bound and has no

effect on the $\lambda$-grid. Therefore, we use the grid proposed by Tolstikhin and Seldin. Namely, we take

$$\lambda_i = c_1^{i-1} \frac{2(n-1)}{n} \left( \sqrt{\frac{n-1}{\ln(1/\delta_1)} + 1} + 1 \right)^{-1}$$

for $i \in \{1, \dots, k_\lambda\}$ and

$$k_\lambda = \left\lceil \frac{1}{\ln c_1} \ln \left( \frac{1}{2} \sqrt{\frac{n-1}{\ln(1/\delta_1)} + 1} + \frac{1}{2} \right) \right\rceil.$$

In the experiments we took $c_1 = 1.05$ and $\delta_1 = \delta/2$.

### D.2.3 The $\gamma$ grid for Theorem 15

The parameter $\gamma$ comes from Theorem 9. By taking the first two derivatives we can verify that for a fixed $\rho$ the PAC-Bayes-Bennett bound is convex in $\gamma$ and at the minimum point the optimal value of $\gamma$ satisfies

$$e^{(\gamma_\rho^* b - 1)} \left( \gamma_\rho^* b - 1 \right) = \frac{1}{e} \left( \frac{b^2 \left( \mathrm{KL}(\rho\|\pi) + \ln \frac{1}{\delta_2} \right)}{n \mathbb{E}_\rho[\tilde{\mathbb{V}}(h)]} - 1 \right).$$

Thus, the optimal value of $\gamma$ is given by

$$\gamma_\rho^* = \frac{1}{b} \left( W_0 \left( \frac{1}{e} \left( \frac{b^2 \left( \mathrm{KL}(\rho\|\pi) + \ln \frac{1}{\delta_2} \right)}{n \mathbb{E}_\rho[\tilde{\mathbb{V}}(h)]} - 1 \right) \right) + 1 \right),$$

where $W_0$ is the principal branch of the Lambert W function, which is defined as the inverse of the function $f(x) = xe^x$.

In order to define a grid for $\gamma$ we first determine the relevant range for $\gamma_\rho^*$. We note that the variance $\mathbb{E}_\rho[\tilde{\mathbb{V}}(h)]$ is estimated using Theorem 13, which assumes that the length of the range of the loss $\tilde{\ell}(\cdot, \cdot)$ is $c$. The loss range provides a trivial upper bound on the variance $\mathbb{E}_\rho[\tilde{\mathbb{V}}(h)] \leq \frac{c^2}{4}$. At the same time, we have $\lambda \left( 1 - \frac{\lambda n}{2(n-1)} \right) \leq \frac{n-1}{2n}$ (it is a downward-pointing parabola) and, therefore, the right hand side of the bound in Theorem 13 is at least the value of its second term, which is at least $\frac{2c^2 \ln \frac{1}{\delta_1}}{n-1}$, since $\mathrm{KL}(\rho\|\pi) \geq 0$. Thus, we obtain that the estimate of $\mathbb{E}_\rho[\tilde{\mathbb{V}}(h)]$ is in the range $\left[ \frac{2c^2 \ln \frac{1}{\delta_1}}{n-1}, \frac{c^2}{4} \right]$. We use $V_{\min} = \frac{2c^2 \ln \frac{1}{\delta_1}}{n-1}$ to denote the lower bound of this range.

Since $W_0(\cdot)$ is a monotonically increasing function, $\mathrm{KL}(\rho\|\pi) \geq 0$, and the estimate of $\mathbb{E}_\rho[\tilde{\mathbb{V}}(h)]$ is at most $\frac{c^2}{4}$, we obtain that $\gamma_\rho^*$ satisfies

$$\gamma_\rho^* = \frac{1}{b} \left( W_0 \left( \frac{1}{e} \left( \frac{b^2 \left( \mathrm{KL}(\rho\|\pi) + \ln \frac{1}{\delta_2} \right)}{n \mathbb{E}_{\rho^2}[\tilde{\mathbb{V}}(h)]} - 1 \right) \right) + 1 \right)$$

$$\geq \frac{1}{b} \left( W_0 \left( \frac{1}{e} \left( \frac{4b^2}{nc^2} \ln \frac{1}{\delta_2} - 1 \right) \right) + 1 \right) \overset{def}{=} \gamma_{\min}.$$

For an upper bound we observe that since $\mathbb{E}_\rho[\tilde{L}(h)] - \mathbb{E}_\rho[\hat{\tilde{L}}(h, S)]$ is trivially bounded by $b$, the bound in Theorem 13 is only interesting if it is smaller than $b$ and, in particular, $\frac{\phi(\gamma b)}{\gamma b^2} \mathbb{E}_\rho[\tilde{\mathbb{V}}(h)] \leq b$. This gives

$$b \geq \frac{\phi(\gamma b)}{\gamma b^2} \mathbb{E}_\rho[\tilde{\mathbb{V}}(h)] \geq \frac{\phi(\gamma b)}{\gamma b^2} V_{\min}.$$

Thus, $\gamma$ should satisfy

$$\phi(\gamma b) \leq \frac{\gamma b^3}{V_{\min}},$$

which gives that the maximal value of $\gamma$, denoted $\gamma_{max}$, is the positive root of

$$H(\gamma) = e^{\gamma b} - \gamma b \left( 1 + \frac{b^2}{V_{\min}} \right) - 1 = 0.$$

Let $\alpha = \left( 1 + b^2/V_{\min} \right)^{-1} \in (0, 1)$, and $x = -\gamma b - \alpha$. Then the above problem is equivalent to finding the root of $f(x) = xe^x - d$ for $d = -\alpha e^{-\alpha}$, which can again be solved by applying the Lambert W function. Since for $\alpha \in (0, 1)$, we have $d \in (-1/e, 0)$, which indicates that there are two roots [Corless et al., 1996]. We denote the root greater than $-1$ as $W_0(d)$ and the root less than $-1$ as $W_{-1}(d)$. It is obvious that $W_0(d) = -\alpha$. However, $W_0(d)$ is not the desired solution, since for $b > 0$, $x = -\alpha$ implies $\gamma = 0$, but we assume $\gamma > 0$. Hence, $W_{-1}(d)$ is the desired root, which gives the corresponding $\gamma = -\frac{1}{b}(W_{-1}(d) + \alpha) > 0$. Thus, we obtain

$$\gamma_{max} = -\frac{1}{b} \left( W_{-1} \left( -\frac{1}{1 + \frac{b^2}{V_{\min}}} \cdot e^{-\frac{1}{1 + \frac{b^2}{V_{\min}}}} \right) + \frac{1}{1 + \frac{b^2}{V_{\min}}} \right).$$

We construct the grid by taking $\gamma_i = c_2^{i-1}\gamma_{\min}$ for $i \in \{1, \dots, k_\gamma\}$, were $k_\gamma = \lceil \ln(\gamma_{max}/\gamma_{\min})/\ln c_2 \rceil$. In the experiments we took $c_2 = 1.05$, and $\delta_1 = \delta_2 = \delta/2$.

### D.2.4 Minimization of the bound

The adjustment of the bound in Theorem 15 to our hypothesis space construction, as described above, is:

$$L(\mathrm{MV}_\rho) \leq \frac{1}{(0.5 - \mu)^2} \left( \mathbb{E}_{\rho^2}[\hat{L}_\mu(h, h', S_h \cap S_{h'})] + \frac{2\,\mathrm{KL}(\rho\|\pi) + \ln\frac{2k}{\delta}}{\gamma n} \right.$$

$$\left. + \frac{\phi(\gamma K_\mu)}{\gamma K_\mu^2} \left( \frac{\mathbb{E}_{\rho^2}[\hat{\mathbb{V}}_\mu(h, h', S_h \cap S_{h'})]}{1 - \frac{\lambda n}{2(n-1)}} + \frac{K_\mu^2 \left( 2\,\mathrm{KL}(\rho\|\pi) + \ln\frac{2k}{\delta} \right)}{n\lambda \left( 1 - \frac{\lambda n}{2(n-1)} \right)} \right) \right),$$

where $n = \min_{h,h'} |S_h \cap S_{h'}|$ and $k = k_\lambda k_\gamma$. We minimize the bound without considering $k_\gamma$ and $k_\lambda$ since we define the grid without taking them into consideration. However, we put back $k_\gamma$ and $k_\lambda$ when computing the generalization bound. Thus, when doing the optimization we take $k = 1$, but when we compute the bound we take the proper $k = k_\lambda k_\gamma$.

---
**Algorithm 2:** Minimization of the bound in Theorem 15
---

    **Input:** $n$, grid of $\mu$ and losses $\mathbb{1}(h(X_i) \neq Y_i)$ for all $(X_i, Y_i) \in S_h$ for all $h$
    **for** $\mu$ selected by the binary search in the grid **do**
        **Initialize:** $\rho = \pi$
        Compute $\hat{L}_\mu(h, h', S_h \cap S_{h'})$ and $\hat{\mathbb{V}}_\mu(h, h', S_h \cap S_{h'})$ for all $h, h'$
        **while** The improvement of the bound for a fixed $\mu$ is larger than $10^{-9}$ **do**
            Compute $\lambda_{\mu,\rho}^*$, the optimal $\lambda$ given $\rho$ and $\mu$
            Compute $\gamma_{\mu,\rho}^*$, the optimal $\gamma$ given $\rho$ and $\mu$
            Apply gradient descent to the bound w.r.t. $\rho$ given $\mu$, $\lambda_{\mu,\rho}^*$ and $\gamma_{\mu,\rho}^*$
        **end while**
        Proceed to the next $\mu$ in the grid proposed by the binary search
    **end for**

---

**Optimal $\lambda$ given $\mu$ and $\rho$**     Given $\mu$ and $\rho$, $\lambda$ can be computed in the same way as in the optimization of Theorem 13, since the optimization problem is the same, and get

$$\lambda_{\mu,\rho}^* = \frac{2(n-1)}{n} \left( \sqrt{\frac{2(n-1)\mathbb{E}_{\rho^2}[\hat{\mathbb{V}}_\mu(h, h', S_h \cap S_{h'})]}{K_\mu^2(2\,\mathrm{KL}(\rho\|\pi) + \ln\frac{2k}{\delta})} + 1} + 1 \right)^{-1}.$$

In our implementation at every optimization step we took the closest $\lambda$ to the above value from the $\lambda$-grid.

**Optimal $\gamma$ given $\mu$ and $\rho$**   Given $\mu$ and $\rho$, the bound for the variance is obtained by plugging in the optimal $\lambda^*_{\mu,\rho}$ computed above. Let

$$U_{\mathbb{V}}(\rho,\mu) = \frac{\mathbb{E}_{\rho^2}[\hat{\mathbb{V}}_\mu(h,h',S_h \cap S_{h'})]}{1 - \frac{\lambda^*_{\mu,\rho} n}{2(n-1)}} + \frac{K_\mu^2 \left(2\,\mathrm{KL}(\rho\|\pi) + \ln\frac{2k}{\delta}\right)}{n\lambda^*_{\mu,\rho}\left(1 - \frac{\lambda^*_{\mu,\rho} n}{2(n-1)}\right)}.$$

Then

$$\gamma^*_{\mu,\rho} = \frac{1}{K_\mu}\left(W_0\left(\frac{1}{e}\left(\frac{K_\mu^2\left(2\,\mathrm{KL}(\rho\|\pi) + \ln\frac{2k}{\delta}\right)}{nU_{\mathbb{V}}(\rho,\mu)} - 1\right)\right) + 1\right),$$

where $W_0$ is the principal branch of the Lambert W function, which is defined as the inverse of the function $f(x) = xe^x$. In our implementation at every optimization step we took the closest $\gamma$ to the above value from the $\gamma$-grid.

**Gradient w.r.t. $\rho$ given $\lambda$, $\gamma$, and $\mu$**   Optimizing the bound w.r.t. $\rho$ is equivalent to constrained optimization of $f(\rho) = \mathbb{E}_{\rho^2}[\hat{L}_\mu(h,h',S')] + a\mathbb{E}_{\rho^2}[\hat{V}_\mu(h,h',S')] + 2b\,\mathrm{KL}(\rho\|\pi)$, where

$$a = \frac{\phi(K_\mu\gamma)}{K_\mu^2\gamma}\frac{1}{1 - \frac{n\lambda}{2(n-1)}}, \quad b = \frac{1}{\gamma n} + \frac{\phi(K_\mu\gamma)}{K_\mu^2\gamma}\frac{K_\mu^2}{n\lambda(1 - \frac{n\lambda}{2(n-1)})},$$

and the constraint is that $\rho$ must be a probability distribution. We optimize $\rho$ in the same way as presented in Appendix D.1. We use $\hat{L}_\mu$ to denote the matrix of empirical $\mu$-tandem losses and $\hat{\mathbb{V}}_\mu$ to denote the matrix of empirical variance of the $\mu$-tandem losses. Then, the gradient w.r.t. $\rho$ is given by:

$$(\nabla f)_h = 2\left(\sum_{h'}\rho(h')(\hat{L}_\mu(h,h',S') + a\hat{\mathbb{V}}\mu(h,h',S')) + b\left(1 + \ln\frac{\rho(h)}{\pi(h)}\right)\right),$$
$$\nabla f = 2\left(\hat{L}_\mu\rho + a\hat{\mathbb{V}}_\mu\rho + b\left(1 + \ln\frac{\rho}{\pi}\right)\right).$$

We applied gradient descent in the same way as presented in Appendix D.1.

# E   Experiments

## E.1   Data sets

As mentioned, we considered data sets from the UCI and LibSVM repositories [Dua and Graff, 2019, Chang and Lin, 2011], as well as Fashion-MNIST (Fashion) from Zalando Research[3]. We used data sets with size $3000 \leq N \leq 70000$ and dimension $d \leq 1000$. These relatively large data sets were chosen in order to provide meaningful bounds in the standard bagging setting, where individual trees are trained on $n = 0.8N$ randomly subsampled points with replacement and the size of the overlap of out-of-bag sets is roughly $n/9$. An overview of the data sets is given in Table 1.

For all experiments, we removed patterns with missing entries and made a stratified split of the data set. For data sets with a training and a test set (SVMGuide1, Splice, Adult, w1a, MNIST, Shuttle, Pendigits, Protein, SatImage, USPS) we combined the training and test sets and shuffled the entire set before splitting.

## E.2   Optimized weighted random forest

**Experimental Setting**

This section describes in detail the settings and the results of the empirical evaluation using random forest (RF) majority vote classifiers.

We construct the ensemble from decision trees available in *scikit-learn*. For each data set, an ensemble of 100 trees is trained using bagging (as described in Section 6). For each tree, the Gini criterion is used for splitting and $\sqrt{d}$ features are considered in each split.

---

[3]`https://research.zalando.com/welcome/mission/research-projects/fashion-mnist/`

Table 1: Data set overview. $c_{\min}$ and $c_{\max}$ denote the minimum and maximum class frequency.

| Data set | $N$ | $d$ | $c$ | $c_{\min}$ | $c_{\max}$ | Source |
|---|---|---|---|---|---|---|
| Adult | 32561 | 123 | 2 | 0.2408 | 0.7592 | LIBSVM (a1a) |
| Cod-RNA | 59535 | 8 | 2 | 0.3333 | 0.6667 | LIBSVM |
| Connect-4 | 67557 | 126 | 3 | 0.0955 | 0.6583 | LIBSVM |
| Fashion | 70000 | 784 | 10 | 0.1000 | 0.1000 | Zalando Research |
| Letter | 20000 | 16 | 26 | 0.0367 | 0.0406 | UCI |
| MNIST | 70000 | 780 | 10 | 0.0902 | 0.1125 | LIBSVM |
| Mushroom | 8124 | 22 | 2 | 0.4820 | 0.5180 | LIBSVM |
| Pendigits | 10992 | 16 | 10 | 0.0960 | 0.1041 | LIBSVM |
| Phishing | 11055 | 68 | 2 | 0.4431 | 0.5569 | LIBSVM |
| Protein | 24387 | 357 | 3 | 0.2153 | 0.4638 | LIBSVM |
| SVMGuide1 | 3089 | 4 | 2 | 0.3525 | 0.6475 | LIBSVM |
| SatImage | 6435 | 36 | 6 | 0.0973 | 0.2382 | LIBSVM |
| Sensorless | 58509 | 48 | 11 | 0.0909 | 0.0909 | LIBSVM |
| Shuttle | 58000 | 9 | 7 | 0.0002 | 0.7860 | LIBSVM |
| Splice | 3175 | 60 | 2 | 0.4809 | 0.5191 | LIBSVM |
| USPS | 9298 | 256 | 10 | 0.0761 | 0.1670 | LIBSVM |
| w1a | 49749 | 300 | 2 | 0.0297 | 0.9703 | LIBSVM |

Table 2: Numerical values of the test loss obtained by the RFs with optimized weighting. The smallest loss is highlighted in **bold**, while the smallest optimized loss is underlined.

| Data set | $L(\mathrm{MV}_{\rho_u})$ | $L(\mathrm{MV}_{\rho_\lambda})$ | $L(\mathrm{MV}_{\rho_{\mathrm{TND}}})$ | $L(\mathrm{MV}_{\rho_{\mathrm{CCTND}}})$ | $L(\mathrm{MV}_{\rho_{\mathrm{CCPBB}}})$ |
|---|---|---|---|---|---|
| SVMGuide1 | **0.0284 (0.0037)** | 0.0372 (0.0066) | 0.0287 (0.0035) | 0.0286 (0.0036) | 0.0287 (0.0039) |
| Phishing | **0.0292 (0.004)** | 0.0371 (0.0073) | **0.0292 (0.0036)** | **0.0292 (0.0036)** | **0.0292 (0.004)** |
| Mushroom | **0.0 (0.0)** | **0.0 (0.0)** | **0.0 (0.0)** | **0.0 (0.0)** | **0.0 (0.0)** |
| Splice | **0.0299 (0.009)** | 0.1087 (0.021) | 0.0306 (0.0099) | 0.0309 (0.0092) | 0.0302 (0.01) |
| w1a | 0.0108 (0.0007) | 0.016 (0.0005) | 0.0108 (0.0006) | **0.0107 (0.0006)** | 0.0108 (0.0006) |
| Cod-RNA | 0.0402 (0.0013) | 0.0712 (0.0064) | **0.0395 (0.0014)** | **0.0395 (0.0014)** | **0.0395 (0.0015)** |
| Adult | **0.1693 (0.0027)** | 0.1942 (0.0151) | 0.1698 (0.0031) | 0.1701 (0.003) | 0.1698 (0.0031) |
| Connect-4 | 0.1706 (0.0023) | 0.2803 (0.0165) | 0.1699 (0.002) | 0.1705 (0.0024) | **0.1695 (0.0019)** |
| Shuttle | **0.0002 (0.0001)** | 0.0003 (0.0002) | **0.0002 (0.0001)** | **0.0002 (0.0001)** | **0.0002 (0.0001)** |
| Pendigits | 0.0096 (0.0023) | 0.0452 (0.0124) | **0.0092 (0.0022)** | 0.0093 (0.0021) | **0.0092 (0.0025)** |
| Letter | **0.0378 (0.0036)** | 0.1408 (0.0356) | 0.0398 (0.0041) | 0.0402 (0.0042) | 0.0383 (0.0034) |
| SatImage | **0.0828 (0.0068)** | 0.1321 (0.0268) | 0.0835 (0.0061) | 0.0839 (0.0062) | 0.0832 (0.006) |
| Sensorless | 0.0014 (0.0004) | 0.0138 (0.0019) | **0.0012 (0.0003)** | **0.0012 (0.0003)** | **0.0012 (0.0003)** |
| USPS | **0.0394 (0.0043)** | 0.1325 (0.0251) | 0.0401 (0.0055) | 0.0405 (0.0052) | 0.0404 (0.005) |
| MNIST | **0.0316 (0.0017)** | 0.16 (0.0352) | 0.0323 (0.0017) | 0.0324 (0.0017) | 0.0317 (0.0014) |
| Fashion | **0.1175 (0.0018)** | 0.2122 (0.0299) | 0.1192 (0.0022) | 0.1197 (0.0022) | 0.1178 (0.0021) |

We compare the RF using the default uniform weighting $\rho_u$ and the optimized weighting obtained by FO [Thiemann et al., 2016], TND [Masegosa et al., 2020], CCTND (Theorem 12) and CCPBB (Theorem 15). Optimization is based on the out-of-bag sets (see Section 6). For each optimized RF, we also compute the optimized bound.

**Numerical Results**

This section lists the numerical results for the empirical evaluation using RF. Table 2 provides the numerical values of the test loss obtained by the RFs with uniform weighting and with weighting optimized by FO, TND, CCTND and CCPBB; a visual presentation is given in Figure 2a. As observed by Masegosa et al. [2020], optimization using FO leads to overfitting, while the second-order bounds does not significantly degrade the performance. Among the second-order bounds, optimizing using CCPBB produces the best classifier in most cases.

Table 3 provides the numerical values of the optimized bounds; a visual presentation is given in Figure 2b. Table 4 provides the recorded Gibbs loss and tandem loss using the optimized $\rho$. The optimal $\mu$ found is reported for CCTND and CCPBB as well.

Table 3: Numerical values of the bounds for the RFs with optimized weighting. The tightest bound is highlighted in **bold**, while the tightest second-order bound is underlined.

| Data set | FO($\rho_\lambda$) | TND($\rho_{\text{TND}}$) | CCTND($\rho_{\text{CCTND}}$) | CCPBB($\rho_{\text{CCPBB}}$) |
|---|---|---|---|---|
| SVMGuide1 | **0.1079 (0.0079)** | 0.1836 (0.0062) | 0.1853 (0.0059) | 0.2806 (0.0071) |
| Phishing | **0.1189 (0.0035)** | 0.1642 (0.0043) | 0.1674 (0.0042) | 0.2336 (0.005) |
| Mushroom | **0.0068 (0.0001)** | 0.0353 (0.0002) | 0.0388 (0.0002) | 0.1121 (0.0006) |
| Splice | **0.3245 (0.0218)** | 0.4077 (0.0062) | 0.4247 (0.0065) | 0.6562 (0.0056) |
| w1a | **0.0424 (0.0015)** | 0.0633 (0.0009) | 0.0642 (0.0009) | 0.0805 (0.0011) |
| Cod-RNA | **0.1629 (0.0018)** | 0.1663 (0.0014) | 0.1698 (0.0014) | 0.19 (0.0018) |
| Adult | **0.4388 (0.0042)** | 0.5701 (0.0051) | 0.5508 (0.004) | 0.5976 (0.0042) |
| Connect-4 | **0.5978 (0.0067)** | 0.6831 (0.0039) | 0.6758 (0.0036) | 0.7112 (0.0038) |
| Shuttle | **0.0026 (0.0002)** | 0.0078 (0.0002) | 0.0083 (0.0002) | 0.018 (0.0003) |
| Pendigits | **0.142 (0.0035)** | 0.1445 (0.0026) | 0.1504 (0.0042) | 0.2155 (0.003) |
| Letter | **0.3858 (0.0067)** | 0.4504 (0.0032) | 0.4513 (0.003) | 0.5134 (0.0039) |
| SatImage | **0.3762 (0.0075)** | 0.4902 (0.0079) | 0.4851 (0.007) | 0.6158 (0.0083) |
| Sensorless | 0.0348 (0.0031) | **0.0257 (0.0006)** | 0.0265 (0.0006) | 0.0376 (0.0007) |
| USPS | **0.3394 (0.0065)** | 0.4059 (0.0048) | 0.4097 (0.0044) | 0.5086 (0.0042) |
| MNIST | 0.3795 (0.0031) | **0.3537 (0.0014)** | 0.3598 (0.0014) | 0.3853 (0.0014) |
| Fashion | **0.4806 (0.003)** | 0.5436 (0.0023) | 0.5408 (0.0021) | 0.5728 (0.0021) |

Table 4: Numerical values for Gibbs loss, tandem loss and optimized $\mu$ for the RFs with optimized weighting. We use $\mathbb{E}_\rho[L]$ and $\mathbb{E}_{\rho^2}[L]$ as short-hands for the Gibbs and the tandem loss respectively.

| Data set | FO $\mathbb{E}_\rho[L]$ | FO $\mathbb{E}_{\rho^2}[L]$ | TND $\mathbb{E}_\rho[L]$ | TND $\mathbb{E}_{\rho^2}[L]$ | CCTND $\mathbb{E}_\rho[L]$ | CCTND $\mathbb{E}_{\rho^2}[L]$ | CCTND $\mu$ | CCPBB $\mathbb{E}_\rho[L]$ | CCPBB $\mathbb{E}_{\rho^2}[L]$ | CCPBB $\mu$ |
|---|---|---|---|---|---|---|---|---|---|---|
| SVMGuide1 | 0.0325 | 0.0217 | 0.0406 | 0.0185 | 0.0403 | 0.0184 | -0.0527 | 0.0413 | 0.0194 | -0.0258 |
| Phishing | 0.041 | 0.0255 | 0.0486 | 0.0197 | 0.0484 | 0.0196 | -0.0295 | 0.049 | 0.0202 | -0.0125 |
| Mushroom | 0.0 | 0.0 | 0.0002 | 0.0 | 0.0002 | 0.0 | -0.0317 | 0.0002 | 0.0 | -0.01 |
| Splice | 0.1068 | 0.0903 | 0.1564 | 0.0424 | 0.1522 | 0.0415 | -0.057 | 0.16 | 0.044 | 0.0045 |
| w1a | 0.0156 | 0.0123 | 0.0179 | 0.0091 | 0.0179 | 0.009 | -0.0111 | 0.018 | 0.0092 | -0.0065 |
| Cod-RNA | 0.0712 | 0.0602 | 0.0802 | 0.0314 | 0.0803 | 0.0314 | -0.0178 | 0.0815 | 0.0318 | 0.0102 |
| Adult | 0.1995 | 0.1474 | 0.2061 | 0.1184 | 0.2056 | 0.1182 | -0.1216 | 0.2068 | 0.1194 | -0.0918 |
| Connect-4 | 0.2824 | 0.2564 | 0.2953 | 0.1523 | 0.2943 | 0.1521 | -0.0959 | 0.2974 | 0.1535 | -0.0615 |
| Shuttle | 0.0003 | 0.0001 | 0.0006 | 0.0002 | 0.0006 | 0.0002 | -0.0044 | 0.0006 | 0.0002 | 0.0 |
| Pendigits | 0.0502 | 0.0346 | 0.061 | 0.0163 | 0.0609 | 0.0163 | -0.0099 | 0.0614 | 0.0166 | 0.0092 |
| Letter | 0.1685 | 0.1249 | 0.1803 | 0.0851 | 0.1797 | 0.0849 | -0.0501 | 0.1816 | 0.0861 | -0.0228 |
| SatImage | 0.1478 | 0.0968 | 0.1612 | 0.0746 | 0.1602 | 0.0741 | -0.1104 | 0.1617 | 0.0755 | -0.0535 |
| Sensorless | 0.0125 | 0.0113 | 0.0192 | 0.0027 | 0.0192 | 0.0027 | 0.0008 | 0.0195 | 0.0027 | 0.01 |
| USPS | 0.1363 | 0.0989 | 0.1517 | 0.0644 | 0.1509 | 0.0641 | -0.053 | 0.1522 | 0.065 | -0.0173 |
| MNIST | 0.1763 | 0.1286 | 0.1837 | 0.075 | 0.1835 | 0.075 | 0.0281 | 0.185 | 0.0756 | 0.037 |
| Fashion | 0.2256 | 0.1715 | 0.2325 | 0.1196 | 0.2322 | 0.1195 | -0.0577 | 0.2334 | 0.1203 | -0.0382 |

## E.3 Ensemble of multiple heterogeneous classifiers

**Experimental Setting**

This section describes in detail the settings and the results of the experimental evaluation using an ensemble of multiple heterogeneous classifiers.

The ensemble is defined by a set of standard classifiers available in *scikit-learn*:

- **Linear Discriminant Analysis**, with default parameters, which includes a singular value decomposition solver.
- Three versions of **k-Nearest Neighbors**: (i) k=3 and uniform weights (i.e., all points in each neighborhood are weighted equally); (ii) k=5 and uniform weights; and (iii) k=5 where points are weighted by the inverse of their distance. In all cases, it is employed the Euclidean distance.
- **Decision Tree**, with default parameters, which includes Gini criterion for splitting and no maximum depth.
- **Logistic Regression**, with default parameters, which includes L2 penalization.

Table 5: Numerical values of the test loss obtained by ensembles of multiple heterogeneous classifiers with optimized weighting. The smallest loss is highlighted in **bold**, while the smallest optimized loss is underlined.

| Data set | $L(\mathrm{MV}_u)$ | $L(h_{best})$ | $L(\mathrm{MV}_{\rho_\lambda})$ | $L(\mathrm{MV}_{\rho_{\mathrm{TND}}})$ | $L(\mathrm{MV}_{\rho_{\mathrm{CCTND}}})$ | $L(\mathrm{MV}_{\rho_{\mathrm{CCPBB}}})$ |
|---|---|---|---|---|---|---|
| SVMGuide1 | 0.0357 (0.005) | 0.0404 (0.0047) | 0.0404 (0.0047) | 0.0352 (0.0051) | 0.0348 (0.0053) | **0.0343 (0.0059)** |
| Phishing | 0.0353 (0.0035) | 0.0459 (0.0058) | 0.0459 (0.0058) | **0.0333 (0.0031)** | 0.0337 (0.0028) | 0.0335 (0.0032) |
| Mushroom | 0.0001 (0.0002) | 0.0002 (0.0004) | **0.0 (0.0)** | 0.0001 (0.0002) | 0.0001 (0.0002) | 0.0001 (0.0002) |
| Splice | 0.1055 (0.0104) | 0.0768 (0.0098) | 0.0768 (0.0098) | 0.075 (0.0093) | 0.0768 (0.0098) | **0.069 (0.0082)** |
| w1a | **0.0125 (0.0007)** | 0.0153 (0.0009) | 0.0153 (0.0009) | 0.0128 (0.0008) | 0.0129 (0.0008) | 0.0128 (0.0007) |
| Cod-RNA | 0.0707 (0.0022) | 0.064 (0.0022) | 0.064 (0.0022) | 0.0552 (0.002) | **0.0551 (0.0019)** | 0.0581 (0.0023) |
| Adult | 0.1627 (0.0036) | 0.1543 (0.0039) | 0.1543 (0.0039) | 0.1563 (0.0042) | **0.1541 (0.0039)** | 0.1566 (0.0048) |
| Protein | 0.3491 (0.0066) | 0.3251 (0.0061) | 0.3251 (0.0061) | **0.3176 (0.0052)** | 0.3251 (0.0061) | 0.3185 (0.0048) |
| Connect-4 | 0.2039 (0.0035) | 0.2433 (0.0032) | 0.2433 (0.0032) | **0.1989 (0.003)** | 0.1992 (0.0032) | 0.2018 (0.0037) |
| Shuttle | 0.0012 (0.0002) | **0.0005 (0.0002)** | **0.0005 (0.0002)** | 0.0006 (0.0002) | 0.0006 (0.0002) | 0.0006 (0.0002) |
| Pendigits | 0.0111 (0.0016) | 0.0092 (0.0017) | 0.0092 (0.0017) | 0.0086 (0.0016) | 0.0087 (0.0016) | **0.0085 (0.0019)** |
| Letter | 0.069 (0.0041) | 0.0673 (0.0052) | 0.0673 (0.0052) | 0.0538 (0.0043) | 0.054 (0.0043) | **0.0526 (0.0041)** |
| SatImage | 0.0997 (0.0069) | 0.1054 (0.0046) | 0.1053 (0.0046) | 0.0939 (0.0061) | 0.0954 (0.0063) | **0.093 (0.0059)** |
| Sensorless | 0.1816 (0.0121) | **0.0213 (0.0018)** | **0.0213 (0.0018)** | **0.0213 (0.0018)** | 0.1089 (0.2764) | **0.0213 (0.0018)** |
| USPS | 0.0359 (0.0054) | 0.0375 (0.0038) | 0.0375 (0.0038) | **0.0324 (0.0044)** | 0.0326 (0.0042) | 0.0326 (0.0036) |
| MNIST | 0.0356 (0.002) | 0.0349 (0.0017) | 0.0349 (0.0017) | **0.0304 (0.0016)** | **0.0304 (0.0017)** | **0.0304 (0.0016)** |
| Fashion | 0.1341 (0.0019) | 0.154 (0.0028) | 0.154 (0.0028) | **0.1323 (0.003)** | 0.1341 (0.003) | 0.1346 (0.0034) |

- **Gaussian Naive Bayes**, with default parameters.

We included three versions of the kNN classifier to test if our bounds could deal with a heterogeneous set of classifiers where some of them are expected to provide highly correlated errors while others are expected to provide much less correlated errors.

Each of the seven classifiers of the ensemble was learned from a bootstrap sample of the training data set. We did it in the way to be able to compute and optimize our bounds with the out-of-bag-samples as described in Section 6.

**Numerical Results**

This section lists the numerical results for the empirical evaluation using ensembles of multiple heterogeneous classifiers.

Table 5 provides the numerical values of the test loss obtained by these ensembles with uniform weighting and with weighting optimized by FO, TND, CCTND and CCPBB; a visual presentation is given in Figure 2c. In this case, uniform voting is not a competitive weighting scheme. The second-order bounds perform much better than uniform weighting and than the weights computed according to the first-order bound. There is not any clear winner among the second-order bounds.

Table 6 provides the numerical values of the optimized bounds; a visual presentation is given in Figure 2d. Among the second-order bounds, the CCTND bound is often tighter in this setting.

Table 7 provides the recorded Gibbs loss and tandem loss using the optimized $\rho$. The optimal $\mu$ found is reported for CCTND and CCPBB as well.

Table 6: Numerical values of the bounds for ensembles of multiple heterogeneous classifiers with optimized weighting. The tightest bound is highlighted in **bold**, while the tightest second-order bound is underlined.

| Data set | $FO(\rho_\lambda)$ | $TND(\rho_{TND})$ | $CCTND(\rho_{CCTND})$ | $CCPBB(\rho_{CCPBB})$ |
|---|---|---|---|---|
| SVMGuide1 | **0.1133 (0.0053)** | 0.221 (0.0127) | 0.2183 (0.0112) | 0.3142 (0.0116) |
| Phishing | **0.1242 (0.0056)** | 0.1957 (0.0075) | 0.1977 (0.0072) | 0.2658 (0.0074) |
| Mushroom | **0.0078 (0.0008)** | 0.0412 (0.0019) | 0.0441 (0.0019) | 0.1162 (0.0026) |
| Splice | **0.2361 (0.0186)** | 0.4772 (0.0286) | 0.4613 (0.0242) | 0.6769 (0.0288) |
| w1a | **0.0392 (0.0015)** | 0.0694 (0.0021) | 0.0703 (0.0021) | 0.0879 (0.0022) |
| Cod-RNA | **0.1448 (0.0026)** | 0.2164 (0.0032) | 0.2148 (0.0031) | 0.2445 (0.003) |
| Adult | **0.3343 (0.0071)** | 0.5648 (0.0077) | 0.5366 (0.0066) | 0.5857 (0.0064) |
| Protein | **0.6944 (0.0057)** | 1.0 (0.0) | 0.9078 (0.0034) | 1.0 (0.0) |
| Connect-4 | **0.5157 (0.0047)** | 0.7272 (0.0099) | 0.6733 (0.0068) | 0.7107 (0.0064) |
| Shuttle | **0.0033 (0.0008)** | 0.0106 (0.0012) | 0.0111 (0.0012) | 0.0215 (0.0011) |
| Pendigits | **0.0335 (0.0033)** | 0.0838 (0.0062) | 0.0856 (0.0061) | 0.1412 (0.0067) |
| Letter | **0.1591 (0.0053)** | 0.2682 (0.0092) | 0.2627 (0.0084) | 0.3154 (0.0099) |
| SatImage | **0.271 (0.0146)** | 0.4908 (0.0123) | 0.4593 (0.011) | 0.5857 (0.0122) |
| Sensorless | **0.0523 (0.0031)** | 0.1173 (0.0057) | 0.2054 (0.2793) | 0.1357 (0.0064) |
| USPS | **0.1069 (0.0053)** | 0.2183 (0.0074) | 0.2142 (0.0066) | 0.2932 (0.0084) |
| MNIST | **0.081 (0.0022)** | 0.139 (0.0049) | 0.1383 (0.0046) | 0.1574 (0.0051) |
| Fashion | **0.3291 (0.0033)** | 0.4945 (0.0066) | 0.4709 (0.0049) | 0.5049 (0.0045) |

Table 7: Numerical values for Gibbs loss, tandem loss and optimized $\mu$ for the heterogeneous classifiers with optimized weighting. We use $\mathbb{E}_\rho[L]$ and $\mathbb{E}_{\rho^2}[L]$ as short-hands for the Gibbs loss and the tandem loss respectively.

| Data set | FO $\mathbb{E}_\rho[L]$ | $\mathbb{E}_{\rho^2}[L]$ | TND $\mathbb{E}_\rho[L]$ | $\mathbb{E}_{\rho^2}[L]$ | CCTND $\mathbb{E}_\rho[L]$ | $\mathbb{E}_{\rho^2}[L]$ | $\mu$ | CCPBB $\mathbb{E}_\rho[L]$ | $\mathbb{E}_{\rho^2}[L]$ | $\mu$ |
|---|---|---|---|---|---|---|---|---|---|---|
| SVMGuide1 | 0.0365 | 0.031 | 0.0457 | 0.026 | 0.0439 | 0.0258 | -0.0691 | 0.046 | 0.0267 | -0.0362 |
| Phishing | 0.0447 | 0.0383 | 0.059 | 0.0262 | 0.0533 | 0.0262 | -0.0419 | 0.059 | 0.0266 | -0.0158 |
| Mushroom | 0.0001 | 0.0 | 0.0041 | 0.0003 | 0.0029 | 0.0002 | -0.0289 | 0.0046 | 0.0003 | -0.008 |
| Splice | 0.0754 | 0.0754 | 0.1265 | 0.0552 | 0.1062 | 0.0548 | -0.19 | 0.1331 | 0.0578 | -0.0672 |
| w1a | 0.0147 | 0.0134 | 0.0204 | 0.0106 | 0.0195 | 0.0106 | -0.0125 | 0.0192 | 0.0108 | -0.0065 |
| Cod-RNA | 0.0638 | 0.0572 | 0.0737 | 0.0425 | 0.0717 | 0.0425 | -0.0356 | 0.0813 | 0.0442 | -0.02 |
| Adult | 0.1502 | 0.1484 | 0.2049 | 0.1181 | 0.1726 | 0.1224 | -0.1771 | 0.1907 | 0.1202 | -0.1168 |
| Protein | 0.3224 | 0.3153 | 0.4052 | 0.2645 | 0.324 | 0.3017 | -1.2391 | 0.4182 | 0.267 | -0.5 |
| Connect-4 | 0.2438 | 0.236 | 0.2612 | 0.1634 | 0.2534 | 0.1638 | -0.221 | 0.2628 | 0.165 | -0.1912 |
| Shuttle | 0.0005 | 0.0004 | 0.0019 | 0.0004 | 0.0014 | 0.0004 | -0.0047 | 0.0021 | 0.0004 | 0.0 |
| Pendigits | 0.008 | 0.0069 | 0.0156 | 0.0062 | 0.0133 | 0.0061 | -0.0299 | 0.0144 | 0.0063 | -0.0118 |
| Letter | 0.0644 | 0.0587 | 0.0768 | 0.0454 | 0.0723 | 0.0454 | -0.0641 | 0.086 | 0.0466 | -0.0407 |
| SatImage | 0.1032 | 0.0928 | 0.1246 | 0.0766 | 0.1188 | 0.0764 | -0.1647 | 0.1284 | 0.0783 | -0.0995 |
| Sensorless | 0.0208 | 0.0208 | 0.0345 | 0.0198 | 0.1161 | 0.1036 | -0.027 | 0.0445 | 0.0202 | -0.0122 |
| USPS | 0.036 | 0.0326 | 0.0478 | 0.028 | 0.0434 | 0.0278 | -0.0676 | 0.0496 | 0.0288 | -0.0357 |
| MNIST | 0.0345 | 0.0304 | 0.0428 | 0.026 | 0.0403 | 0.026 | -0.0272 | 0.0482 | 0.0265 | -0.017 |
| Fashion | 0.1528 | 0.1488 | 0.1665 | 0.1081 | 0.1636 | 0.1084 | -0.1173 | 0.1724 | 0.1092 | -0.094 |