# OpenReview forum: "Chebyshev-Cantelli PAC-Bayes-Bennett Inequality for the Weighted Majority Vote"
_NeurIPS.cc/2021/Conference — NeurIPS 2021 Poster_

### Official Review · Reviewer_AxQU · 2021-07-14

**Rating:** 9
**Confidence:** 5

**Summary:**

This submission is about bounding the risk of the weighted majority vote classifier formed by aggregating the predictions of a set of base classifiers according to a certain weighting scheme (a distribution over hypotheses). It is motivated by the observation that previous bounds were loose or based on restrictive assumptions, and may not be convenient for optimization when they cannot be converted to empirical bounds that could be used as learning objectives in view of the fact that they depend on inaccessible oracle quantities.

Specifically, the submission proves a parametric version of the Chebyshev-Cantelly inequality (Theorem 6), which is used as the basis to develop bounds on the risk of the $\rho$-weighted majority vote (Theorem 7, Theorem 8), where the upper bounds depend on the expected tandem loss of pairs of hypotheses chosen randomly according to $\rho$ (case of Theorem 8) and additionally the expected loss of the randomized classifier that chooses single hypotheses randomly according to $\rho$ (case of Theorem 7), besides depending on an optimizable parameter $\mu$. The latter two expected losses (tandem and single) are then controlled by using PAC-Bayesian inequalities in order to convert the oracle bounds into empirical bounds. One such empirical bound (on the expected risk of the weighted majority vote),  given in Theorem 12, is developed from Theorem 7 when bounding the $\rho$-expected losses (tandem and single) in terms of their empirical counterparts by means of the PAC-Bayes-$\lambda$ inequalities. Another such empirical bound, given in Theorem 15, is developed from Theorem 8 when bounding the $\rho$-expected loss (single) by a careful combination of building blocks, one key of which is a new inequality (Theorem 9) called PAC-Bayes-Bennett inequality, which is novel to the best of my knowledge, and is one of the main contributions of this submission.

A very interesting feature of these new empirical bounds (Theorem 12 and Theorem 15) is that, besides being valid simultaneously for all "posterior" distributions $\rho$ (as is typical in PAC-Bayes inequalities), they hold simultaneously (i.e. uniformly) over possible values of their parameters, which implies that the parameters can be selected by optimization without paying the classical union bound price. A set of experimental results then show the benefits of using the new bounds for learning weighted majority votes as well as computing bounds on their risk. The experiments also give comparisons of the new bounds to previous bounds in the relevant literature.

**Limitations And Societal Impact:**

Sure.

**Main Review:**

ORIGINALITY

The contributions of this submission are novel, to the best of my knowledge. The "parametric Chebyshev-Cantelly" (Theorem 6) has a surprisingly simple proof, I would be surprised if nobody had come across this inequality in any area of probability or mathematics (though it is possible), but in any case this inequality is used as the basis for deriving two novel oracle bounds for the risk of the weighted majority vote, hence there is novelty in its use, regardless of the inequality itself. Another novel contribution of this submission, as pointed above in my summary (and visibly emphasized in the abstract and other places) is the "PAC-Bayes-Bennett" inequality, which is an improvement on the "PAC-Bayes-Bernstein" inequality (and detailed analytical comparison between the two is given) and is the basis for converting an oracle bound into an empirical bound, besides possibly being interesting by itself.

SIGNIFICANCE

The theoretical sections are interesting for the machine/statistical learning theory community, and the experimental demonstrations are additionally interesting for practitioners and anyone interested in machine learning with guarantees. There is sufficient informative discussion of the analytical properties of the contributed inequalities and the convenient properties of the new empirical bounds. The relevant literature is discussed and the previous works are cited with fair attributions. The section on experimental results demonstrates the improved tightness of the computed bounds with respect to previous related results (without compromising the test set errors as far as I understood), and the presentation of the experimental setting and discussion of results is overall a positive example that I would love to see followed by other works.

CLARITY / QUALITY / READABILITY

Reading this paper was a pleasure. No exaggeration. I consider this submission to be a positive example of good scientific writing: the narrative is clear and grammatically correct mostly (minor blemishes pointed below), the material is organized in a coherent way so that following it required no unreasonable efforts, and the choice of mathematical presentation (notations, derivations) is likewise so that one can follow/verify the arguments without requiring unreasonable efforts; my compliments for this. The proofs look correct. Overall, this submission is high quality in my opinion.

EDITORIAL FEEDBACK

L.8: "derive a new inequality that we call PAC-Bayes-Bennett inequality"
L.10: Replace "by" with "of"
L.11: Replace "by" with "of"

L.24-25: Replace "weight tuning" with "tuning the weights" (or with a hyphen "weight-tuning" I think?)

L.38: "held-out"

L.43: Is is possible to mention what are the restrictive settings? Say writing "restrictive settings (e.g. [restriction 1] and [restriction 2])" if space allows to mention them briefly.
L.44: "in their translation to" (i.e. insert "their")

L.66: "We derive a new inequality that we name the PAC-Bayes-Bennet inequality, which [...]"
L.67: Replace "by" with "of"

L.96: "defined the \emph{tandem loss}" (i.e. insert "the")

L.97: Preferably write \emph{joint error} or "joint error" (in quotations) for emphasis

L.109: "the variance of a random variable $Z$"

L.137 (Theorem 7):  Should say "In multiclass classification, for any $\mu < 0.5$" (I think)

L.152 (Theorem 8): Same comment.

L.155-156: "we can use Bernstein-type inequalities [Seldin et al. 2012] to obtain tighter estimates compared to kl-type inequalities [Seeger 2002, see Theorem 10 below]."

L.168: "fixed (but arbitrary) $\gamma>0$"

L.172: Replace "by" with "of" (and same in L.173 and in L.175)

L.197: Perhaps clarify after the theorem that the upper bound (4) is due to Thiemann et al. (2017), while the lower bound (5) follows by a similar argument as noted by Masegosa et al. (2020).

Theorem 12: Should the $\mu$ values in the grid be all $\leq 0.5$? since the argument is based on Theorem 7, which requires $\mu \leq 0.5$ I think. Also, in the upper bound, replace the outermost parenthesis with square brackets for better visibility. The first KL in the upper bound has a factor of 2 that I am not sure where it comes from. If this is a typo, it's easy to correct (delete the factor 2). If this is no typo, it calls for commenting in the proof outline (L.208-210) why the factor 2 is present. The proof outline could me explicit saying that the "result follows by substitution of the upper bound on E_{ρ^2} [L(h, h′)] and the lower bound on E_{ρ}[L(h)] given by Theorem 11 (namely, by (4) and (5), respectively) into Theorem 7"

L.212: Replace "the" with "their" and for clarity specify the possible range, for instance write
"their range (e.g. $0, 1$ vs. $-1, 1$) and expectation"

L.225-226: I suggest to reduce the last sentence to "Rescaling to a general range introduces a new factor."
Then state Theorem 13 and after the theorem statement comment on the new factor $c^2$.

Theorem 13: "with range c" is ambiguous --- please clarify according to the intended meaning. By "range" I understand the set of possible values that this loss may take, but clarify if you mean something different.

L.231: Since the PAC-Bayes-kl inequality does not involve any $\gamma$ parameter, the meaning of "in contrast to the PAC-Bayes-kl inequality" is unclear.

L.235: This use of the word "range" is unusual, could you clarify?
The set of possible values of the $\mu$-tandem loss is { $\mu^2$, $\mu^2-\mu$, $(1-\mu)^2$ } and this is what I understand as range. These values are according to whether the losses of $h$ and $h'$ are both 0, or one is 0 while the other is 1, or both are 1.
On a quick look at Appendix B, it looks that you are conflating "range" (as I understand it) with "length of the range" so this needs to be sorted out. For instance, say if a function's range (in mathematical meaning) is the interval [1, 5], then this range has length 5-1 = 4. I think the quantity $K_{\mu}$ in your Lemma 14 corresponds to "length" as just described.

L.239 (Theorem 15): Give explicitly the restrictions (if any) on the $\mu$'s and the $\gamma$'s and the $\lambda$'s. At least the $\mu$'s should be restricted to be $\leq 0.5$ I think.

L.242-245: Please also comment on the factor 2 in front of the two KL terms in the bound. It is not clear to me why these factors 2 should be present, but feel free to convince me that these factors are correct. In case if these factors should not be present, I would request that the experimental results that rely on Theorem 15 should be revised. [Similarly, any experimental results relying on Theorem 12 should be revised if the factor 2 on the first KL should be removed. Double check to be sure.]

L.311: Replace "supplementary material" with "Appendix E" (and do the same in L.316)

L.324: Replace "by" with "of"

L.326: "search in parametric bounds, which would be cleaner and could give tighter values than a union bound over a grid"

Only skimmed the appendices but am willing to look at the details in the appendices when we get there.

**Time Spent Reviewing:**

5 hours

---

> ### Author Response · Authors · 2021-08-09
> **Rebuttal**
>
> We thank the reviewer for extremely detailed feedback and editorial comments.
>
> The grids in Theorems 12 and 15: Yes, for the grids we have that all the values of $\mu$ are smaller than 0.5 and, in the case of Theorem 15, all the values of $\lambda$ are in the $(0,1]$ interval and all the values of $\gamma$ are positive.
>
> Factor 2 in front of the KL in Theorems 12 and 15: we have $KL(\rho^2||\pi^2) = 2 KL(\rho||\pi)$, see Line 202 for an explanation. We will add this to the proofs.
>
> Range: we are very sorry about the confusion in terminology. The “range” in Theorem 13 and Lemma 14 means the “length of the range”, as you have explained. We will write this clearer.

---

### Official Review · Reviewer_WkCd · 2021-07-15

**Rating:** 7
**Confidence:** 4

**Summary:**

This paper provides new second order oracle bounds for the weighted majority vote classifier. The classical second order bounds (C-bound) has a variance term in the denominator which makes it harder to estimate and optimize the term empirically. Recently, a different second order bound was proposed which did not have these issues and was thus, easier to optimize and estimate but is somewhat looser compared to the C-bound. This paper gives an intermediate bound which is tighter than the recent bound but also easier to optimize and estimate. This paper also provides a new concentration bound in the PAC Bayesian setting - PAC-Bayes-Bennett inequality to relate the term in the oracle bound to its empirical value.

**Limitations And Societal Impact:**

Not applicable.

**Main Review:**

The weighted majority vote is a very well used classifier and getting tight generalization bounds which can also be used to get better weighting schemes is important and would be relevant to other members of the community as well.
The bounds are the application of a different second order bound which in my opinion is novel. Although the application does not seem technically challenging, the results look promising to me and also lead to better performance experimentally which is interesting.

The paper is very well written and easy to follow. The comparison to previous works is very clear.

I see that the optimized weightings are not compared to the C-bounds. The paper mentions that the optimization of that bound is hard but are there existing works which try to optimize based on these bounds and did the paper try to compare using those methods?

One thing is that the authors have not discussed the comparison of their two bounds. The authors note on line 300 that one of their bounds leads to better accuracy when the weights are optimized whereas the other bound leads to tighter generalization bound. This seems puzzling. Can the authors comment more on this observation? Also, do the authors have insight into which one of their bounds is better or worse under what conditions? I think it would be good to mention this in the paper to get a better understanding.

**Time Spent Reviewing:**

3

---

> ### Author Response · Authors · 2021-08-09
> **Rebuttal**
>
> We thank the reviewer for their time and supportive feedback.
>
> Comparison to C-bounds: we are unaware of computationally efficient ways to optimize the prior C-bounds, this is why a comparison is impossible.
>
> Comparison of the two bounds (Theorem 12 and Theorem 15): it is hard to make a concise comparison of the relative power of the bounds, because of multiple factors pushing in different directions. The tightness of Theorem 15 depends on a good choice of the parameter grid and on whether the empirical $\mu$-tandem loss has low variance, which also depends on the optimal value of $\mu$. The variance of $\mu$-tandem loss depends on the interplay between the data and the hypothesis set. The parameter grid of Theorem 12 involves just one parameter, so it has relatively little influence, but Theorem 12 cannot exploit the low variance of the tandem loss.

---

### Official Review · Reviewer_ViFo · 2021-07-15

**Rating:** 7
**Confidence:** 3

**Summary:**

This paper worked on the analysis of the majority vote based on a second-order bound. The proposed analysis is based on the extension of the Chebyshev-Cantelli inequality, which uses the offset. The new bound is tighter or more optimization-friendly than the existing bound. With this new bound, the authors proposed a new majority vote algorithm and applied it to the random forest and heterogeneous classifiers. The numerical experiments support the usefulness of the proposed method especially when heterogeneous classifiers are used.

**Limitations And Societal Impact:**

Yes, they are shown in the checklist.

**Main Review:**

Overall, the paper provides a practical and rigorous extension for the analysis of the majority vote. I think this is good enough to publish.

# Pros
- The motivation and goal of this work are explicitly written in Section 1 and one can quickly understand why this work is important. Also, section 3 is well written concisely, and easy to follow for a non-expert.

# Cons
-  The algorithm for deriving the optimal $\rho$,  which is required for numerical experiments,  is not shown even in Appendix. I thought the authors should summarize Appendix D into the algorithm.

# Comments and Questions:
- In Figure 1, around the area of $E_{\rho}L(h) \approx 0.5$ and $E_{\rho ^2}L(h,h')  \approx 0.5$, theorem 7 is tighter than theorem 3. I would like to know an intuition about this. That area means the high correlation of errors and each hypothesis does not work well since $E_{\rho}L(h)\approx 0.5$. Also, on the area, $E_{\rho}L(h) = E_{\rho ^2}L(h,h')$, theorem 7 seems tightest because it means the numerator of Theorem 5 is zero. What is the intuition of this case?

- In Figure 2, in experiments a) and b), using the optimized $\rho$ for a random forest model seems not to work well in all the methods (TND, C$\mu$TND, COTND). On the other hand, the application to heterogeneous classifiers worked well. Why this happened and is this explainable from any existing or proposed bound ? Is this caused by the low correlations of errors of heterogeneous classifiers ?

- In Figure 2, especially, c) and d), there are gaps regarding the behavior of loss and bound. For example, in splice experiments, in Fig c) COTND shows the smallest loss, while in Fig d) COTND shows the largest bound. The authors argued that this is caused by the union bound in the derivation. I would like to know the reason why this gap is caused in more detail.

- This is just a minor comment, why author named the bound in Theorem 12 as C$\mu$TND and the bound in Theorem 15 as COTND ? (abbreviation ?)

**Time Spent Reviewing:**

4  hours

---

> ### Author Response · Authors · 2021-08-09
> **Rebuttal**
>
> We thank the reviewer for their time and supportive feedback.
>
> We will provide a pseudo-code for deriving the optimal $\rho$.
>
> Figure 1: for comparison and intuition it is easier to consider Theorem 5, which, as an oracle bound, is equivalent to Theorem 7. Also for simplicity, assume that all hypotheses have the same expected loss, i.e., $L(h)$ is the same for all $h$. If $L(h,h’) = L(h)$ for all pairs $h, h’$, it means that all predictions are perfectly correlated and $E_\rho[L(h)] = E_{\rho^2}[L(h,h’)]$. In this case Theorem 3 gives $L(MV_\rho) \leq 4 E_\rho^2[L(h,h’)] = 4 L(h)$, whereas Theorem 5 gives $L(MV_\rho) \leq 4 (E_\rho^2[L(h,h’)] - E_\rho[L(h)]^2) = 4(L(h) - L(h)^2) = 4 L(h) (1-L(h))$, which explains why Theorem 5 is tighter than Theorem 3 and why the advantage grows as $L(h)$ approaches 0.5. As we mention in the comment to Figure 1 (Lines 262-263), in this regime the first order oracle bound is tighter than both second order bounds, so this is not a very interesting region, but we still improve on Theorem 3.
>
> Note that the numerator of Theorem 5 is zero when $E_\rho[L(h)]^2 = E_{\rho^2}[L(h,h’)]$, not when $E_\rho[L(h)] = E_{\rho^2}[L(h,h’)]$ (note the square on the LHS). To get some intuition regarding the implications of the former equality, consider the case where all hypotheses have the same first order loss $L(h)$, the number of hypotheses is large, and $\rho$ is uniform. The expected tandem loss of $h$ and $h’$ can be expressed as: $L(h, h’) = P(h(x) \neq y)P(h’(x)  \neq  y | h(x)  \neq y)$. Under the above assumptions (and being a bit sloppy with notation for brevity),  we have $E_\rho[L(h)] = P(h(x) \neq y)$ (we can use any $h$ on the right hand side, since they all have the same loss), and $E_{\rho^2}[L(h,h’)] \approx P(h(x) \neq y)P(h’(x)  \neq  y | h(x)  \neq y)$ (assuming the number of hypotheses is large and that $P(h’(x)  \neq  y | h(x)  \neq y)$ is roughly the same for all pairs $h,h’$, so we can use any $h$ and $h’$ on the right hand side).  So, if $E_\rho[L(h)]^2 = E_{\rho^2}[L(h,h’)]$, we have that $P(h'(x)  \neq  y | h(x)  \neq y)\approx P(h(x) \neq y)$, which means that the errors of $h$ and $h’$ are roughly independent. Thus, in case of independent errors, identical first order losses, and uniform weighting, the C-bound converges to zero (this is discussed by Germain et al. (2015)), whereas the bound of Masegosa et al. based on the second order Markov’s (Theorem 3 in our paper) does not (this is discussed by Masegosa et al.).
>
>
> Figure 2: yes, in our experiments with heterogeneous classifiers the correlation in their prediction errors was lower than in the case of random forest and, therefore, the improvement relative to the tandem bound was more significant. This is quite expected, because in random forest all predictors are of the same type (decision trees), and so the correlation of errors is higher.
>
> Figure 2: the term $\ln(2 k_\mu k_\lambda k_\gamma / \delta)$ coming from the union bound is quite large and pushes the COTND bound up in Figure 2.d. It is hard to comment on individual down-spikes in Figure 2.c, because they could be due to statistical fluctuations.
>
> Naming: C$\mu$TND is named so, because it is a C-bound based on the tandem loss (TND) and using a parameter $\mu$. COTND is a C-TND bound with an Offset (the O stands for Offset).
>
> After reflecting on your comment we decided to change the names to CCTND for Theorem 12 (Chebyshve-Cantelli bound with TND empirical loss estimate) and CCPBB for Theorem 15 (Chebyshev-Cantelli bound with PAC-Bayes-Bennett loss estimate). We will add the explanation to the paper. If you have further feedback regarding the naming, we are open to your suggestions.

---

### Official Review · Reviewer_Am83 · 2021-07-16

**Rating:** 7
**Confidence:** 2

**Summary:**

 The paper aims at developing tools for analyzing majority-based  learners. It does this through the idea of oracle bounds. The central  and known idea, is to bound the error using the oracle error and then  bound the oracle error by a function of the empirical error. The authors  introduce new bounds for the two stages of the process, thereby  improving the known bounds in the field.


**Limitations And Societal Impact:**

The authors indeed explain the limitations of the results and the regimes on which they do and do not obtain improvement over the current result in the area.

**Main Review:**

The article is well written. It is clear and well-organized. The results are presented in a sequential and properly constructed manner. In addition, the results are compared to previous results along the way, providing context and perspective. The practical implications of the theoretical results are also properly explained, and the experiments adequately illustrate this.
Overall, the results appear effective and interesting. The improvement that these results achieve beyond those already known, is limited, but it exists. Additionally, the authors present tools that may be useful in other contexts or may produce additional results in the current field of research.

**Time Spent Reviewing:**

10

---

> ### Author Response · Authors · 2021-08-09
> **Rebuttal**
>
> We thank the reviewer for their time and supportive feedback.

---

### Official Review · Reviewer_hg22 · 2021-07-17

**Rating:** 5
**Confidence:** 3

**Summary:**

This paper addresses the problem of bounding the generalization error of weighted majority vote classifiers.  They begin with oracle-inequalities, and show how to convert them to inequalities in terms of fully empirical quantities, without substantially weakening the bounds. In particular, to bound to the 0-1 loss, they introduce an alternative to the Chebyshev-Cantelli inequality, which promises to be easier to optimize and apply than the $C$-bound.  Furthermore, to bound tandem losses they introduce a PAC-Bayes-Bennett inequality, which improves over the PAC-Bayes-Bernstein inequality in the manner one would expect.


**Main Review:**


This paper addresses several concerns in the computation and optimization of bounds of this nature, in particular the Chebyshev-Cantelli bound. An issue I have is that, while the oracle-based PAC-Bayes inequalities operate nonuniformly over the entire space of possible reweightings in a manner that seems quite competitive with a union bound approach, once the oracles are dropped and fully empirical bounds are adopted, it seems that they do not beat a union-bound approach, except in very particular circumstances, which are not clearly outlined and do not seem reasonable to this reviewer. Furthermore, much of the work seems quite incremental; e.g., simply replacing Chebyshev-Cantelli with Chebyshev and Bernstein with Bennett.  Overall, while there are some interesting ideas and nice results, these issues detract from the work as a whole and don't leave me convinced of its utility. If the empirical bounds could be improved, and particularly if the grid over possible parameters could be removed with an argument that centers the grid over some true optimal parameterization, I would be inclined to change my opinion of this work.


My major issue is with the fully empirical inequalities 10 and 11.  The elegance, beauty, and utility of the PAC-Bayes inequalities is that they give a bound for all possible re-weighting simultaneously, without paying a union-bound type price in the numerator of $\ln(1/\delta)$.  An obvious (and obviously flawed) alternative would be to upper-bound all $d^2$ tandem losses simultaneously via the union bound, introducing $\ln \frac{d^{2}}{\delta}$ terms.  Unfortunately in section 5, theorems 10 and 11, the lovely $\ln \frac{1}{\delta}$ terms become $\ln \frac{\sqrt{n}}{\delta}$ terms. Direct comparison to the union bound is a bit difficult, since everything is still in terms of $\lambda$ and $\gamma$ variables, which must be optimized over, but with these sub-gamma and sub-Poisson inequalities, we should expect to see these logarithmic terms appear fast decaying range-based (non-square root) and slow decaying variance-based (square root) terms.

I thus remain unconvinced that the results of this section beat the obvious union bound strategy, unless $n \leq d^{4}$.  This analysis holds, even not accounting for the KL divergence terms, or the need to establish a grid, so it does tell us the size of the grid $k$ clearly must obey $k \in o(d^{2}/\sqrt{n})$.  Furthermore, since everything is inside a logarithm and presumably $n$ grows faster than $d$, it really seems like the benefit is negligible, and the cost is $O(\log(n))$.

It would be very interesting to see a comparison to [1], which treats a similar problem with such a union bound approach, in particular comparing to their suggestion of using uniform convergence methods, i.e., Rademacher averages, to estimate appropriate statistics.

Specifics:




As I understand it, binary weighted majority vote is a type of linear classification, so why are methods for linear classification not compared against or discussed?

35 Isn't it the lack of correlation of errors that gives the majority vote its power?

75/76: Often $L$ is called risk, written $R$ to distinguish from $\ell$, also called loss.

89: Is $\leq$ here just due to ties?

104: $h$ on LHS is a weighted combination, $h$ on RHS is a random variable?

165: Please specify that $\pi$ is a distribution over voters (unless I’m wrong)?

190: PAC-Bayes-kl -> PAC-Bayes-KL?

196: I'm not sure it matters, but I feel like the $\forall \lambda, \gamma$ should be before the “for all distributions simultaneously”. Also, do (4) and (5) hold individually with probability $1 - \delta$, or simultaneously?

202: $\rho^{2}, \pi^{2}$ are independent product distributions?



Theorem 13 does appear to be based on a sub-gamma bound on the variance. I wonder if it would be possible to get a sub-Poisson bound on the variance? This would compose nicely with the sub-Poisson of the Bennett inequality; otherwise it seems to be a bottleneck?


References:

“Adversarial Multi Class Learning under Weak Supervision with Performance Guarantees,” Mazzetto, Alessio and Cousins, Cyrus and Sam, Dylan and Bach, Stephen H and Upfal, Eli


**Time Spent Reviewing:**

9

---

> ### Author Response · Authors · 2021-08-09
> **Rebuttal**
>
> We believe that most of the criticism comes from a misunderstanding of the results. Below we explain the main flaws of the arguments in the review, most of which are factually incorrect statements.
>
> First, regarding the difference between Chebyshev-Cantelli (a.k.a. one-sided Chebyshev’s) and Chebyshev’s (a.k.a. two-sided Chebyshev’s) inequalities.
>
> Chebyshev-Cantelli inequality states:
> $P(Z - E[Z] \geq \alpha) \leq Var[Z] / (\alpha^2 + Var[Z])$.
>
> Chebyshev’s inequality states:
> $P(|Z - E[Z]| \geq \alpha) \leq Var[Z] / \alpha^2$.
>
> If $\alpha$ is close to zero, Chebyshev-Cantelli can be arbitrarily tighter than Chebyshev. It is not true that “the improvement never exceeds a factor 2”, as stated in the review.
>
> In our case we bound uncentered random variables, i.e., we use
> $P(Z \geq \epsilon) = P(Z - E[Z] \geq \epsilon - E[Z]) \leq Var[Z]/((\epsilon - E[Z])^2 + Var[Z])$.
> In other words, we have $\alpha = \epsilon - E[Z]$ with $\epsilon = 0.5$, and if $E[Z]$ is close to 0.5, then $\alpha$ is close to zero and the improvement is arbitrarily large.
>
> The statement that we “essentially rederive Chebyshev’s inequality” is also incorrect. We rederive the Chebyshev-Cantelli inequality. Substitution of the optimal value of $\mu$, which is $\mu = E[Z] - Var[Z] / (\epsilon - E[Z])$, into Theorem 6, yields Chebyshev-Cantelli inequality. One of our main contributions is that by writing the Chebyshev-Cantelli inequality in a different way we obtain a bound that has no variance in the denominator and, thus, is easy to optimize and to estimate from data. And, at the same time, we keep the tightness of Chebyshev-Cantelli.
>
> The comparison of the PAC-Bayesian approach with a plain union bound is also flawed. If comparing $\ln(d^2/\delta)$ term with $\ln(\sqrt{n}/\delta)$ term, the latter is tighter whenever $d \geq n^{1/4}$ and not $d \geq n^2$, as stated by the reviewer.
>
> Theorem 10 is due to Seeger (2002) (a bit more precisely, Seeger has $\ln((n+1)/\delta)$ term and Maurer (2004) improves it to $\ln(2\sqrt{n}/\delta)$). It is one of the most basic and commonly used results in the PAC-Bayesian literature and the $\ln(2\sqrt{n}/\delta)$ term is standard there. We did not understand the criticism of using this theorem in our work. If the reviewer is aware of a tighter bound, we will appreciate getting a reference.
>
> “... binary weighted majority vote is a type of linear classification …” - this statement is also incorrect: binary weighted majority vote clearly produces non-linear decision boundaries.
>
> “35 Isn’t it the lack of correlation of errors that gives the majority vote its power?” - this is correct. The first order method takes neither correlations nor lack of correlations into account.
>
> “89 …” - yes, $\leq$ is due to ties.
>
> “104 …” - $h$ on both sides is a random variable.
>
> “165 …” - $\pi$ is a distribution on $\mathcal{H}$, where $\mathcal{H}$ is a set of voters.
>
> “190 …” - for two Bernoulli random variables with biases $p$ and $q$, $kl(p||q)$ is a shortcut for $KL([1-p,p]||[1-q,q])$, which is the Kullback-Leibler divergence between the two Bernoulli distributions. The correct name is PAC-Baes-kl, as we wrote. It was given by Seldin et al. (2012) and refers to the small kl on the left-hand side of the inequality. The small kl provides a measure of distance between the empirical and expected loss of the corresponding Gibbs classifiers.
>
> “196 …” - it is for all $\rho, \lambda$, and $\gamma$, so the order does not matter. (4) and (5) hold simultaneously with probability at least $1-\delta$, we will clarify that.
>
> “202: $\rho^2, \pi^2$ are independent product distributions?” - $\rho^2$ and $\pi^2$ are product distributions. We do not understand what kind of “independence” the question refers to.
>
> “Theorem 13 …” - Bennett’s inequality is a sub-Gaussian inequality, not a sub-Poisson. See, for example, Boucheron et al. (2013).

---

> > ### Comment · Reviewer_hg22 · 2021-08-20
> > **Reviewer Response**
> >
> > Thank you for the detailed response.  I was in error regarding lemma 6, and it seems my other criticisms were not shared by the remaining reviewers, so I shall raise my score accordingly. That said, I still have some questions concerns and responses.
> >
> > *Regarding the two-factor gap:*
> >
> > Apologies this statement was supremely unclear.
> > What I should have said is that either the Chebyshev can tell about is greater than $\frac{1}{2}$, in which case topologically it can't be improved by more than a factor $2$, as all probabilities are $\leq 1$, or the Chebyshev-Cantelli bound is less than $\frac{1}{2}$, in which case it does hold that the Chebyshev bound is no more than a factor 2 looser than the Chebyshev-Cantelli.
> > To be precise, we have $ \min(1, Var[Z] / (\alpha^2 + Var[Z])) \leq 2 Var[Z] / (\alpha^2 + Var[Z])$.
> >
> >
> > I don't consider this a breaking issue, but it is an important limitation worth discussing, in particular since you place a major emphasis on your result as a variant of the Chebyshev-Cantelli bound that removes variance from the denominator. As I see it, this is true, but it is only a constant factor improvement relative to the Chebyshev bound (when the above is taken into account).
> >
> >
> > *Regarding lemma 6*
> >
> > As for lemma 6, I was in error. I must have mixed up the notations somewhere, because I thought here that $\mu = \mathbb{E}[Z]$, which yields a one-tailed Chebyshev.  Of course this does not make sense, as $\mu$ is selected and optimized over later. I sincerely apologize for this inaccurate criticism.
> >
> >
> > *Regarding the linear classifiers*
> >
> > “... binary weighted majority vote is a type of linear classification …” - this statement is also incorrect: binary weighted majority vote clearly produces non-linear decision boundaries.
> >
> > I meant that the decision boundary is linear in the *voting space* i.e., the hypercube of joint votes in the binary case. Clearly the statement doesn't make sense over the original domain, where the concept of linearity may be nonsensical.
> > I thought this was worth comparing to, because the classic Vapnik-Chervonenkis bounds for linear classifiers also obtain an $O(\log(n))$ term (like the PAC-Bayesian bounds), although modern methods are able to remove this.
> >
> >
> > *PAC Bayesian bounds*
> >
> > Yes, I believe $n \leq d^{4}$ is the cutoff, thank you. This is slightly better, but I still don't see how it beats the union bound for large sample sizes. Furthermore, this comparison is extremely favorable to your method, because it's completely ignoring the KL term.  The union bound would give a guarantee dependent only on variances for every possible weighting, whereas your bounds are centered on some $\pi$, and thus necessarily stronger when $KL(\rho || \pi)$ is small.
> > I don't have a way of removing the $\sqrt{n}$ cost, my question is, rather, can you argue this cost is worthwhile in light of the union bound analysis (over each expected loss and tandem loss)?  Is there a reason to think we're in the regime where this is inconsequential, or is there some reason that $d$ should be growing appropriately quickly with $n$?
> >
> > *Additional comments on theorem 12*
> >
> > Initially I thought that the $k$ term in Theorem 12 made this comparison to the union bound substantially worse, but upon further reflection, I actually do not think the union bound is necessary.
> > Theorem 7 should hold for any choice of $\mu$ deterministically (and thus for all $\mu$ simultaneously), and Theorem 11 does not depend on $\mu$, so can the $4k\sqrt{n}$ terms be improved to $4\sqrt{n}$?  Similarly, in theorem 15, while $k_{\gamma}$ and $k_{\lambda}$ appear necessary with this argument, can $k_{\mu}$ be removed?
> >
> > *Sub-Poisson vs Sub-Gaussian*
> >
> > I assume your reference refers to the passage “On the other hand, if $v$ is the dominant term in the denominator of the exponent, Bennett’s and Bernstein’s inequalities are almost equivalent, and both provide a sub-Gaussian type inequality.”
> >
> > This is an excellent text, but I can only surmise that the authors were speaking loosely here.  On the previous page, they state “Recall from Section 2.2 that the upper bound is just the logarithm of the moment generating function of a centered Poisson random variable with parameter $v$.  Therefore, the Cramér transform of $S$ is also bounded by that of a corresponding Poisson random variable.”
> >
> > If bounding the MGF by a Poisson MGF is not the definition of sub-Poisson, then I don't know what is. Yes it is true that they are closely related, but if we consider how the bound behaves as a function of sample size, the initial (range-sensitive) fast decay followed by slow (variance-sensitive) decay is characteristic of sub-Poisson (or sub-gamma for that matter), whereas sub-Gaussian bounds (like the Hoeffding bound) generally decay at a uniform slow rate.

---

> > > ### Author Response · Authors · 2021-08-24
> > > **Rebuttal - 2**
> > >
> > > We thank the reviewer for revising the score. Below we provide further explanations to the additional comments.
> > >
> > > Factor-two gap:
> > >
> > > We assume there was a typo in the comment and it referred to $\min(1, Var[Z]/\alpha^2) \leq 2 Var[Z]/(\alpha^2+Var[Z])$, which means that Chebyshev truncated at 1 is at most twice Chebyshev-Cantelli. This is true. Still, a factor 2 difference is very significant if we are looking for tight and practically relevant generalization bounds.
> > >
> > > Regarding the linear classifiers:
> > >
> > > VC bounds in this context are not praised for their tightness and we are unaware of any prior work where they empirically delivered non-trivial generalization guarantees (smaller than 1).
> > >
> > > The $d^4 \geq n$ cut-off:
> > >
> > > For $d=10$ the cut-off is $n=10,000$. For $d=100$ the cut-off is $n=100,000,000$. For most realistic examples we can think of, we are in the regime where PAC-Bayes has an advantage over the plain union bound.
> > >
> > > Additional comments on theorem 12:
> > >
> > > The optimal value of $\mu$ depends on empirical quantities and, therefore, the union bound cannot be dropped. It is somewhat similar to the difference between the Hoeffding’s inequality and PAC-Bayes-Hoeffding inequality. In Hoeffding’s inequality the optimal value of $\lambda$ is independent of the data and no union bound is required, whereas in PAC-Bayes-Hoeffding the optimal value of $\lambda$ depends on the data and a union bound is required (see Seldin et al. (2012) “PAC-Bayesian inequalities for martingales.”)
> > >
> > > Sub-Poisson vs Sub-Gaussian:
> > >
> > > The inequality in Theorem 13 provides fast convergence rates, see Theorem 3 by Tolstikhin and Seldin (2013) for an alternative way of writing the theorem, where this is directly visible.

---

### Decision · Program_Chairs · 2021-09-28

**Decision:**

Accept (Poster)

**Comment:**

The referees are in agreement that this submission provides novel techniques for analyzing majority-based learners. It is very much within the conference scope and of sufficient interest and novelty. All of the referee objections have been addressed during the discussion phase.

**Consistency Experiment:**

NeurIPS has a long history of experimentation. In 2014, NeurIPS ran an experiment in which 10% of submissions were reviewed by two independent committees to quantify the randomness in the review process. This year, we repeated a variant of this experiment to see how the quality of the review process has changed over time.  This paper was part of the experiment and was therefore assigned to two committees (consisting of reviewers, an Area Chair, and a Senior Area Chair) that reached independent decisions.  If both committees made the same recommendation, this recommendation was followed. If a single committee recommended acceptance, the paper was accepted (with the exception of a few cases in which the other committee identified what we considered a fatal flaw, e.g., an error in a key result).

This copy’s committee reached the following decision: **Accept (Poster)**

The other committee assigned to the paper recommended **Reject**.  You can find the other set of reviews, along with any follow up discussion with the authors here:
https://openreview.net/forum?id=_HRYvHFgHeV